

# An aldehyde as a rapid source of secondary aerosol precursors: Theoretical and experimental study of hexanal autoxidation

Shawon Barua[1,*], Siddharth Iyer[1], Avinash Kumar[1], Prasenjit Seal[1], and Matti Rissanen[1,*]

[1]Aerosol Physics Laboratory, Physics Unit, Faculty of Engineering and Natural Sciences, Tampere University, 33720 Tampere, Finland

**Correspondence:** Shawon Barua (shawon.barua@tuni.fi) and Matti Rissanen (matti.rissanen@tuni.fi)

**Abstract.** Aldehydes are common constituents of natural and polluted atmospheres, and their gas-phase oxidation has recently been reported to yield highly oxygenated organic molecules (HOM) that are key players in the formation of atmospheric aerosol. However, insights into the molecular level mechanism of this oxidation reaction have been scarce. While OH initiated oxidation of small aldehydes, with two to five carbon atoms, under high $NO_x$ conditions generally leads to fragmentation products, longer chain aldehydes involving an initial non-aldehydic hydrogen abstraction can be a path to molecular functionalization and growth. In this work, we conduct a joint theoretical-experimental analysis of the autoxidation chain reaction of a common aldehyde, hexanal. We computationally study the initial steps of OH oxidation at the RHF-RCCSD(T)-F12a/VDZ-F12//$\omega$B97X-D/aug-cc-pVTZ level, and show that both aldehydic (on C1) and non-aldehydic (on C4) H-abstraction channels contribute to HOM via autoxidation. The oxidation products predominantly form through the H-abstraction from C1 and C4, followed by fast unimolecular 1,6 H-shifts with rate coefficients $1.7 \times 10^{-1}$ $s^{-1}$ and $8.6 \times 10^{-1}$ $s^{-1}$, respectively. Experimental flow reactor measurements at variable reaction times show that hexanal oxidation products including HOM monomers up to $C_6H_{11}O_7$ and accretion products $C_{12}H_{22}O_{9-10}$ form within 3 seconds reaction time. Kinetic modeling simulation including atmospherically relevant precursor concentrations agrees with the experimental results and the expected timescales. Finally, we estimate the hexanal HOM yields up to seven O atoms with mechanistic details through both C1 and C4 channels.

## 1 Introduction

Aldehydes are important compounds in natural and polluted troposphere with typical atmospheric lifetimes on the order of 10 hours or less. They play an important role in the atmosphere as prompt $HO_x$ and $RO_x$ radical sources (Vandenberk and Peeters, 2003). They are commonly formed, or directly emitted, from several biogenic and anthropogenic processes (Lipari et al., 1984). The anthropogenic emissions of many straight-chain aldehydes, essentially from incomplete fossil fuel combustion and biomass burning, are higher than the emissions of *n*-alkanes with the same carbon numbers (Schauer et al., 1999a, b). In the natural environment they are emitted directly by vegetation or are formed during the first steps of photo-oxidation of a multitude of volatile organic compounds (VOC) (Ciccioli et al., 1993; Carlier et al., 1986). They are the common products in the reactions of alkenes with ozone, and are thus prevalent within most biogenic VOC oxidation processes (Calogirou et al., 1999). They are



often toxic, irritate mucous membranes, eyes and skin, and cause odor problems at relatively low concentrations (Vandenberk
and Peeters, 2003; Ernstgård et al., 2006).

On a global scale the fate of aldehydes is predominantly governed by OH-reactions and photolysis during daytime (Calvert
et al., 2011; Mellouki et al., 2003, 2015), while reactions with $NO_3$ radicals are prevalent during the night (Calvert et al., 2011).
Although OH-induced oxidation of aliphatic aldehydes with up to 5 carbon atoms has been extensively studied (Castañeda et al.,
2012; Iuga et al., 2010; Manion et al., 2015; Wang et al., 2015; Albaladejo et al., 2002), research on longer chain aldehydes
is scarce. Due to the weaker bond strength of the aldehydic hydrogen, the oxidation of aldehydes is frequently initiated by the
abstraction of that atom (Calvert et al., 2011; Mellouki et al., 2003, 2015). Under high-$NO_x$ conditions, this commonly leads
to $C_{n-1}$ alkyl nitrates, $C_{n-1}$ aldehydes and $C_{n-1}$ alkoxy isomerization products (Chacon-Madrid et al., 2010) via scission of
the carbon chain (red arrows in Fig. 1) through acyl (i.e., CO loss) and acyloxy (i.e., $CO_2$ loss) intermediates (Rissanen et al.,
2014; Vereecken and Peeters, 2009). The exceptions are the $C_n$ peroxy acids (PA) and $C_n$ peroxyacyl nitrates (PAN) (Calvert
et al., 2011; Mellouki et al., 2003, 2015; Chacon-Madrid et al., 2010) formed in reactions of acyl peroxy radicals (APR)
with $HO_2$ and $NO_2$, respectively (see Fig. 1). The subsequent branching of the $C_{n-1}$ alkoxy radical towards isomerization,
decomposition and reaction with $O_2$ depends on the size and substitution of the alkyl chain, the longer chains ($\geq C_7$) favoring
the isomerization paths (Atkinson and Arey, 2003; Vereecken and Peeters, 2010; Lim and Ziemann, 2005, 2009; Kwok et al.,
1996; Atkinson, 2007).

In the last decade, highly oxygenated organic molecules (HOM) have been identified as large, direct contributors to atmo-
spheric secondary organic aerosol (SOA) (Ehn et al., 2014; Öström et al., 2017; Bianchi et al., 2019; Brean et al., 2019, 2020),
the dominant component of tropospheric fine particulate matter influencing oxidative capacity, local and global air quality, cli-
mate change and human health (Laden et al., 2006; Hallquist et al., 2009; Spracklen et al., 2011; Huang et al., 2014; Jacobson
et al., 2000; Hansen and Sato, 2001; Kanakidou et al., 2005; Zhang et al., 2014). HOM are produced through a sequential
progression of peroxy radical ($RO_2$) hydrogen-shift reactions (i.e., H-shifts) and molecular oxygen additions in a process
called *autoxidation*. In the case of alkenes and aromatics, the process may also proceed via ring closure reactions forming
organic peroxides (Xu et al., 2019; Rissanen et al., 2015; Glowacki et al., 2009; Møller et al., 2020). Autoxidation causes a
rapid increase in the oxygen content of the molecule (Rissanen et al., 2014, 2015; Crounse et al., 2013; Jokinen et al., 2014;
Mentel et al., 2015; Berndt et al., 2015, 2016), even up to three $O_2$ additions in sub-second timescales (Iyer et al., 2021), and
produces progressively functionalized products that can participate in aerosol formation and growth. Aldehydes are common
first-generation products in several hydrocarbon oxidation sequences (Atkinson and Arey, 2003), and thus their tendency to
form HOM is of special interest.

Chacon-Madrid et al. (2010) have studied the SOA yields from several *n*-aldehyde oxidation systems under high $NO_x$
conditions and contrasted the findings with similar *n*-alkane oxidation. Under their high $NO_x$ reaction conditions, they found
significantly lower SOA yields for the aldehydes, and attributed it to the relatively volatile PAN formation (see Fig. 1). In a
more recent work targeting HOM in cyclohexene oxidation, $NO_2$ was found to suppress HOM formation (Rissanen, 2018),
especially the important low volatile dimer products (Tröstl et al., 2016), shedding some light to the low SOA mass observed
by Chacon-Madrid et al. (2010).







**Figure 1.** General reaction mechanism of *n*-aldehyde oxidation by OH radical. The aldehydic H-abstraction channel (C1) leads to fragmentation and isomerization products while a non-aldehydic H-abstraction channel (Cn) leads to isomerization products. The isomerization channels associated with green arrows are continued in Fig. 4. Under high $NO_x$ conditions, the acyl peroxy radical (APR) forms an acyloxy radical (violet arrow), which fragments into a $C_{n-1}$ intermediate (+ $CO_2$), and ultimately to a $C_{n-1}$ terminal alkoxy radical (blue arrow) that can either isomerize or react with $O_2$ to form a $C_{n-1}$ aldehyde. The isomerization channel (orange arrow) is generally favored for long straight chain aldehydes.



Previous mechanistic understanding of gas-phase VOC oxidation states that the abstraction of the aldehydic H-atom by e.g., OH leads rapidly to fragmentation, and is therefore not expected to lead to VOC functionalization. However, recent experimental studies have shown that the abstraction of the aldehydic hydrogen promotes HOM formation (Rissanen et al., 2014; Ehn et al., 2014; Tröstl et al., 2016; Wang et al., 2021), and is thus expected to increase the SOA yields, especially under low $NO_x$ conditions. Thus, in this work, we set out to resolve this apparent discrepancy and study the molecular level gas-phase oxidation mechanism of a common aldehyde by a joint theoretical-experimental approach, focusing on HOM formation by autoxidation. As most of the HOM products detected appear to contain the same number of carbon atoms as the parent VOC (Bianchi et al., 2019), the pathways that do not break the carbon chain are of interest. Branched and substituted aldehydes are more prone to decomposition as alkoxy intermediates derived from these often tend to undergo fragmentation rather than isomerization reactions. In linear aldehydes, the non-fragmentation pathways likely involve H-atom abstraction from a carbon atom that is distant from the aldehydic moiety. Although a major fraction of the H-abstraction occurs from the aldehydic carbon, other abstraction channels are more likely to promote functionalization and become more competitive as the carbon chain size increases.

In this work, we investigate the H-abstraction by OH of hexanal, and the subsequent H-shift chemistry leading to HOM through aldehydic and non-aldehydic H-abstraction channels. Hexanal was chosen as a surrogate for the larger aldehydes because its size was deemed suitable; it is large enough to allow for efficient H-shifts, but not too large to make high-level quantum chemical computations unfeasible. It should be noted that the larger aldehydes and their oxidized products are invariably more complex than the hexanal model system studied here, and the added functional groups can often increase the rate of autoxidation. To the best of our knowledge, this is the first time a detailed autoxidation mechanism of HOM formation from aldehydes is presented, leading to several HOM products from hexanal by a single OH oxidant attack.

## 2 Methods

### 2.1 Quantum chemical calculations

We employ quantum chemical calculations to find the most efficient route to an OH oxidized hexanal product containing 7 oxygen atoms. Some of the intermediates and transition states along the reaction pathway possess RR and RS configurational isomers, each with many potential conformers. Since the inter-conversion between different isomers typically involves the breaking and reformation of covalent bonds, and is consequently associated with high barriers, we consider the RR and RS isomers separately.

### 2.1.1 Conformer Sampling and geometry optimization

Systematic conformer sampling is performed using the Merck Molecular Force Field (MMFF) method implemented in the Spartan′18 program with a neutral charge enforced on the radical center (Wavefunction, 2018; Møller et al., 2016). The con-





formers are generated by varying all torsional angles of each molecular species by 120 degrees (rotating every nonterminal

bond 3 times).

Initial geometry optimizations are performed at the B3LYP/6-31+G* level. For molecules containing three or more O atoms (resulting in more conformers that are also computationally heavy to optimize), the number of conformers is first reduced by performing a single point electronic energy calculation at the B3LYP/6-31+G* level of theory. Conformers with electronic energies within 5 kcal/mol relative to the lowest energy conformer are considered for geometry optimization at the same level of

theory. Subsequently, conformers within 2 kcal/mol in relative electronic energies are optimized at the $\omega$B97X-D/aug-cc-pVTZ level (Chai and Head-Gordon, 2008; Dunning Jr, 1989; Kendall et al., 1992). All calculations following the initial conformer sampling are carried out using the Gaussian 16 program (Frisch et al., 2016).

The transition states (TS) corresponding to specific hydrogen shifts are found by first constraining the H atom at an approximate distance from the relevant C and O atoms (1.3 Å and 1.15 Å, respectively) and optimizing the structure. The optimized

geometry is then used as an input for an unconstrained TS calculation. These calculations are carried out at the B3LYP/6-31+G* level of theory. Once the TS geometry is found, an MMFF conformer sampling step is carried out using Spartan '18 with the O—H and H—C bond lengths constrained. Additionally, partial bonds with torsions enabled are added to these two bonds prior to the conformer sampling step. The partial bonds have been reported to improve the MMFF optimization resulting in geometries that are closer to local energy minima during the conformer sampling (Draper et al., 2019). The resulting TS

conformers from the conformer sampling step are once again optimized, first with the TS relevant bonds constrained, followed by an unconstrained TS optimization, both at the B3LYP/6-31+G* level of theory. Finally, TS geometries within 2 kcal/mol of the lowest energy geometries are optimized at the $\omega$B97X-D/aug-cc-pVTZ level of theory.

Transition states corresponding to the OH H-abstraction of hexanal are found using the same approach, except for the aldehydic H-abstraction, in which case, the initial TS optimization is carried out using the MN15/def2-tzvp level of theory

instead of B3LYP/6-31+G* since the latter method failed to find the TS structure. The conformer sampling step on the OH aldehydic H-abstraction TS structures did not lead to additional conformers.

On the lowest electronic energy reactant, intermediate, TS and product geometries at the $\omega$B97X-D/aug-cc-pVTZ level (MN15/def2-tzvp for the aldehydic H-abstraction case), single-point calculation at the RHF-RCCSD(T)-F12a/VDZ-F12 level is carried out using the Molpro program version 2019.2 (Werner et al., 2019). The T1 diagnostic numbers of all reactants, TS

and products considered here are below 0.03 and 0.045 for closed-shell and open-shell species, respectively, indicating that these systems are single-reference and the CCSD(T) numbers reported here are therefore reliable.

## 2.2   RRKM Calculations

We employ the Master equation solver for multi-energy well reactions (MESMER) program (Glowacki et al., 2012) to carry out RRKM simulations to estimate the branching ratios of selected products following unimolecular isomerization and O$_2$ addition

reactions. The unimolecular isomerization reactions are treated using the SimpleRRKM method with Eckart tunneling. The O$_2$ addition reactions to carbon-centered radicals are treated using the "Simple Bimolecular Sink" method in MESMER. In this case, a bimolecular loss rate coefficient of $2 \times 10^{-12}$ cm$^3$ molecule$^{-1}$ s$^{-1}$ and an O$_2$ "excess reactant" concentration of



$5 \times 10^{18}$ molecules cm$^{-3}$ are used, which is the approximate O$_2$ concentration under standard atmospheric conditions. The intermediates are assigned as "modeled" in the simulations and given Lennard–Jones parameters sigma = 6.25 Å and epsilon

= 343 K (Hippler et al., 1983). MESMER utilizes the exponential down ($\Delta E_{down}$) model for simulating the collisional energy transfer. For N$_2$ bath gas, the MESMER recommended values for $\Delta E_{down}$ are between 175 and 275 cm$^{-1}$. We used a $\Delta E_{down}$ value of 225 cm$^{-1}$ in our simulations. In addition, a grain size of 100 and a value of 60 k$_B$T for the energy spanned by the grains were used. The MESMER input file corresponding to one of the studied reactions is provided in Supplement section S5 as an example.

## 2.3   Rate coefficients

The unimolecular H-shift rate coefficients ($k$) reported in this work are calculated using the multiconformer transition-state theory (MC-TST) (Møller et al., 2016) including quantum mechanical tunneling (Henriksen and Hansen, 2018), as shown in Eq. (1).

$$k = \kappa \frac{k_B T}{h} \frac{\sum_i^{all\,TS\,conf.} \exp\left(-\frac{\Delta E_i}{k_B T}\right) Q_{TS,i}}{\sum_j^{all\,R\,conf.} \exp\left(-\frac{\Delta E_j}{k_B T}\right) Q_{R,j}} \exp\left(-\frac{E_{TS} - E_R}{k_B T}\right) \tag{1}$$

The constants, $k_B$, and $h$ are Boltzmann's constant and Planck's constant, respectively. Absolute temperature, $T$, is set to 298.15 K. $\Delta E_i$ is the zero-point-corrected energy of the $i^{th}$ TS conformer relative to the lowest-energy transition state conformer, and $Q_{TS,i}$ is the partition function of the $i^{th}$ transition state conformer. Similarly, $\Delta E_j$ and $Q_{R,j}$ are the corresponding values for reactant conformer $j$. $E_{TS}$-$E_R$ is the zero-point corrected barrier height corresponding to the lowest energy TS and reactant conformers. The partition functions are calculated at the $\omega$B97X-D/aug-cc-pVTZ level of theory (MN15/def2-tzvp for the

aldehydic H-abstraction case), while the energies include the final coupled-cluster correction. The tunneling coefficient $\kappa$ is calculated using the one-dimensional Eckart approach as reported in Møller et al. (Møller et al., 2016) This method requires the energies of the reactant and product wells that are connected to the lowest energy TS geometry, which are found by running forward and reverse intrinsic reaction coordinate (IRC) calculations on that TS geometry and optimizing the end geometries at B3LYP/6-31+G* level of theory. The reactant and product wells are subsequently re-optimized at the $\omega$B97X-D/aug-cc-pVTZ

level of theory, followed by single-point RHF-RCCSD(T)-F12a/VDZ-F12 energy corrections.

To calculate the rate coefficients of the H-abstraction reactions of hexanal by OH, the bimolecular TST expression shown in Eq. (2) is used.

$$k = \frac{k_B T}{h \cdot P_{ref}} \exp\left(-\frac{G_{TS} - G_R}{k_B T}\right) \tag{2}$$

where, $P_{ref}$ is the total concentration of molecules in standard condition, $2.46 \times 10^{19}$ molecules cm$^{-3}$. $G_{TS}$ and $G_R$ are the

Gibbs free energies (at 298.15 K and 1 atm) of the TS and the reactant, respectively.



## 2.4 Chemical ionization mass spectrometry

The experimental hexanal OH oxidation reaction is conducted in a laboratory applying a flow rector setup (see Fig. 2). A nitrate-based time-of-flight chemical ionization mass spectrometer (nitrate-CIMS) is used to detect the products of hexanal OH oxidation. Chemical ionization is achieved by supplying synthetic air (sheath flow) containing nitric acid ($HNO_3$) under exposure to X-rays. This produces nitrate ($NO_3^-$) ions which are mixed with the sample flow and ionize HOM as $NO_3^-$ adducts. A sheath flow of 20 L min$^{-1}$ and a sample flow of around 10 L min$^{-1}$ (synthetic air as diluent) are used. The precursors are mixed in a quartz flow tube reactor where the oxidant OH is produced in-situ by the ozonolysis reaction of tetramethylethylene (TME) (Berndt and Böge, 2006) (see Fig. 2). An ozone concentration of 225 ppb which is generated by flowing synthetic air through an ozone generator fitted with a 184.9 nm (UVP) Hg Pen-Ray lamp is allowed to react with 40 ppb of TME supplied form a gas cylinder to the reactor. A hexanal precursor concentration of 1 ppm is added to the reactor from another gas cylinder. The reaction time is controlled by adjusting the distance between the mass spectrometer orifice and the place where hexanal and OH meet inside the reactor. This is done by providing the hexanal flow through a moveable injector tube within the reactor. Accordingly, separate sets of experiments are conducted with variable reaction times (1.4, 3.1 and 12 s) to track the oxidation chain propagation. In addition, to confirm the structures of the identified products in favor of the proposed mechanisms, we conduct hydrogen/deuterium (H/D) exchange experiment (10 s) by adding $D_2O$ from a bubbler with $N_2$ flow.

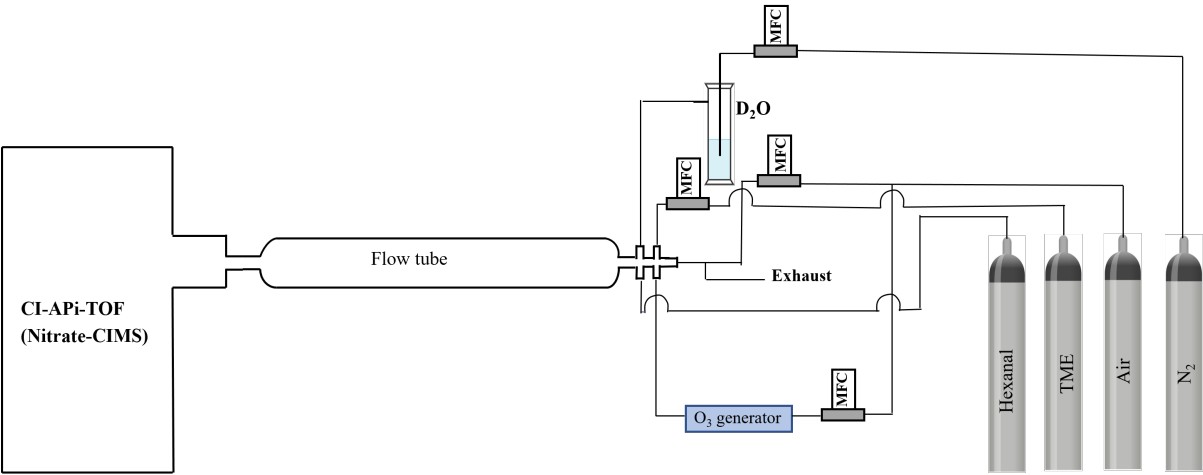

**Figure 2.** Schematic of the flow reactor setup used in the hexanal OH oxidation reaction. The oxidant OH radical is produced in-situ by TME + $O_3$ reaction. TME = tetramethylethylene. All flows are controlled by mass flow controllers (MFC). The flow tube length is adjusted in the variable residence time experiments.



## 2.5 Kinetic modelling simulation

We inspect autoxidation propagation with bimolecular interventions under pristine boreal forest to polluted urban conditions using a kinetic simulator Kinetiscope version 1.1.956.x64 (Hinsberg and Houle, 2017). A single reactor model with constant

volume, pressure, and temperature is employed in the simulation. The temperature is set to 298.15 K. In the simulation setting, a total number of particles $1 \times 10^8$, a random number seed 12947 are used and the maximum simulation time is set to 20 s.

## 3 Results and discussion

The calculated yields of the different H-abstraction channels of hexanal by OH (see Table 1) show that the abstraction from the aldehydic carbon C1 is the dominant channel, which is unsurprising due to the known low bond dissociation enthalpy

of an aldehydic H atom. The next competitive H-abstraction channel is from the secondary carbon atom C4, which has a significantly lower barrier in comparison with the adjacent C3 and C5 sites. All H-abstraction channels with labelled carbon atoms and corresponding relative electronic energies of TSs and products are shown in Fig. 3.

**Table 1.** Overall reaction and TS energies in kcal mol$^{-1}$ of the different OH H-abstraction reactions of hexanal along with calculated rate coefficients (in cm$^3$ molecule$^{-1}$ s$^{-1}$) and branching ratios.

| H-abstraction channels | $E_a$ | $\Delta ZPE$ | $\Delta G^\pm$ | $\Delta G$ | $k^i$ | BR (%) | $k_{SAR}$ |
|---|---|---|---|---|---|---|---|
| C1 (aldehydic H)[b] | -0.03 | -29.1 | 6.9 | -30.1 | $2.14 \times 10^{-12}$ | 92.1 | $2.56 \times 10^{-11}$ |
| C2 ($\alpha$ H) | 2.0 | -26.9 | 9.5 | -27.3 | $2.66 \times 10^{-14}$ | 1.2 | $7.10 \times 10^{-13}$ |
| C3 ($\beta$ H) | 1.5 | -19.2 | 9.2 | -20.6 | $4.76 \times 10^{-14}$ | 2.1 | $1.16 \times 10^{-12}$ |
| C4 ($\gamma$ H) | 0.1 | -19.1 | 8.8 | -20.9 | $9.16 \times 10^{-14}$ | 4.0 | $1.16 \times 10^{-12}$ |
| C5 ($\delta$ H) | 1.6 | -19.9 | 10.3 | -21.3 | $7.35 \times 10^{-15}$ | 0.3 | $9.47 \times 10^{-13}$ |
| C6 (primary H) | 2.7 | -17.1 | 10.3 | -17.9 | $7.27 \times 10^{-15}$ | 0.3 | $1.60 \times 10^{-13}$ |

[b] aldehydic H-abstraction barrier calculated at RHF-RCCSD(T)-F12a/VDZ-F12// MN15/def2-tzvp level of theory. BR = branching ratio. $k_{overall} = 2.32 \times 10^{-12}$ cm$^3$ molecule$^{-1}$ s$^{-1}$. $k_{SAR}$ = structure-activity relationship (SAR) prediction rate coefficients (Jenkin et al., 2018; Ziemann and Atkinson, 2012).

Since the C1 and C4 channels have the highest branching ratios, we focus on these to study the formation of HOM. Nevertheless, the other abstraction channels can also potentially contribute, and could provide additional pathways forming highly

functionalized reaction products. Due to the rapid scaling up of the possible isomerization pathways, and due to the exponential increase in the required computing resources for larger system sizes, we limited our study to the formation of HOM with up to 7 oxygen atoms. These molecules are still radical intermediates that can potentially autoxidize and lead to molecules with even more oxygen atoms.





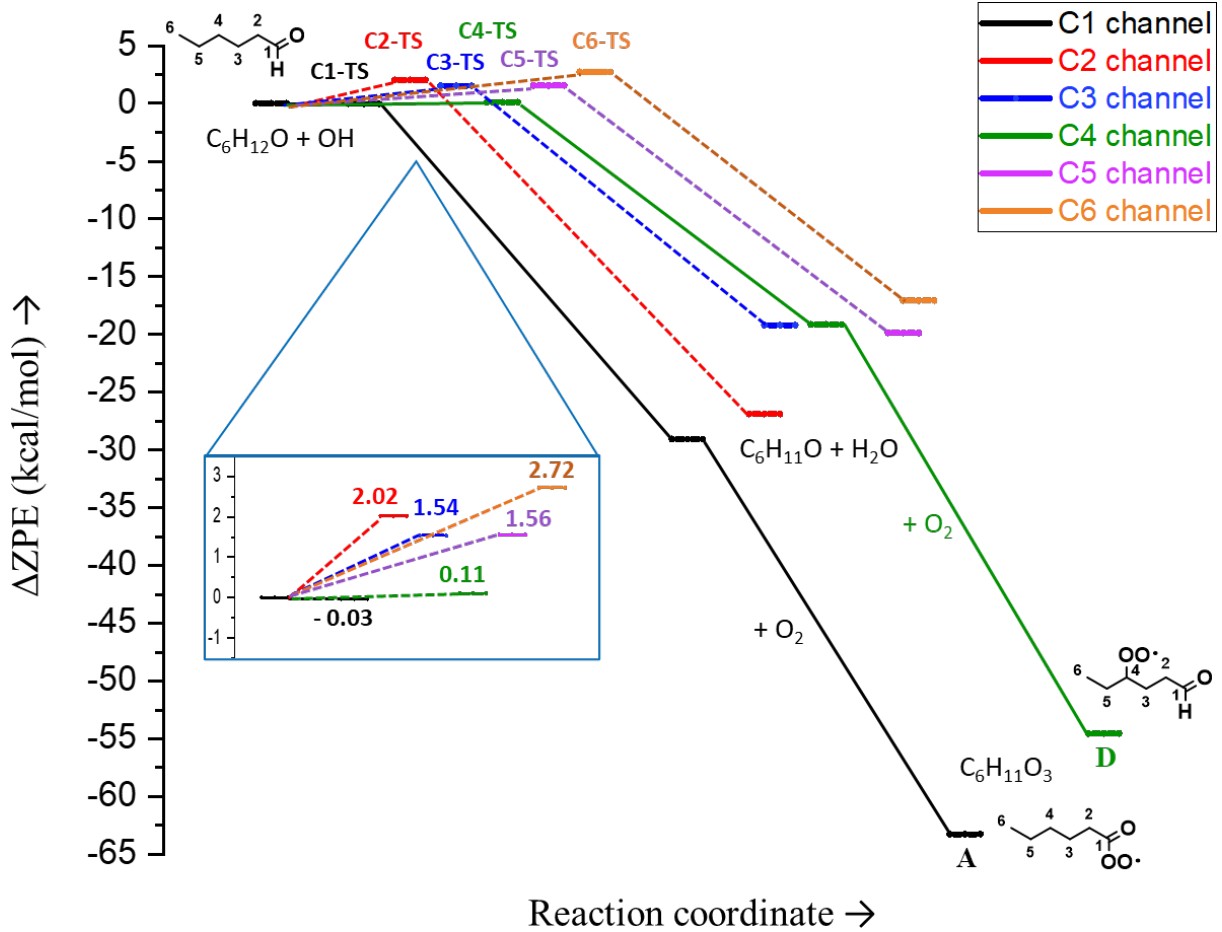

**Figure 3.** Relative electronic energies (zero-point corrected) of TSs and products in different H-abstraction channels of hexanal OH oxidation reaction. The enlarged view of the reaction barriers, $E_a$ ($\Delta ZPE^{\pm}$), is presented in the inset of the figure. The potential energy surface (PES) is extended for C1 and C4 channels up to the intermediates A and D respectively ($C_6H_{11}O_3$).

### 3.1 H-abstraction rates

Relative zero-point corrected electronic energies (ZPE) and Gibbs free energies (for T = 298.15 K and P = 1 atm) of the TSs and overall reactions for the H-atom abstraction reactions of hexanal by OH are reported in Table 1. $E_a$ ($\Delta ZPE^{\pm}$) is the barrier of the reaction, which is calculated as $\Delta ZPE^{\pm} = ZPE_{TS} - ZPE_{Reactants}$; $\Delta ZPE$ is the overall reaction energy, $\Delta ZPE = ZPE_{Products} - ZPE_{Reactants}$. Analogously, $\Delta G^{\pm}$ is the Gibbs free energy of activation, and $\Delta G$ is the reaction Gibbs free energy. The H-abstraction rate coefficients shown in Table 1 were calculated using Eq. (2). The corresponding branching

ratios (BR) are determined as ratios of the individual rate coefficients ($k_i$) against overall rate coefficient ($k_{overall}$).

Ā   The results of Table 1 show that the zero-point corrected electronic TS energy of the aldehydic H-abstraction in hexanal OH oxidation is slightly below the separated reactants, the $\gamma$ H-abstraction on C4 has almost no barrier (i.e., 0.1 kcal mol$^{-1}$), while





the other H-abstraction channels have discernable barriers. The $\beta$ and $\delta$ H-abstractions on C3 and C5, respectively, have near identical barriers, while the $\alpha$ H-abstraction barrier on C2 is 0.7 kcal mol$^{-1}$ less than the primary H-abstraction on C6. The

H-abstraction barriers tend to increase as we move from C4 towards the carbon atoms either along the left or the right direction of the carbon chain (excluding the aldehydic carbon). This trend is consistent with the results of Castañeda et al. (2012), who showed that from ethanal to pentanal OH oxidation the $\beta$ H-abstraction barriers are less than that of $\alpha$ H-abstractions, while the aldehydic H-abstractions have the lowest barriers.

Based on the decreasing trend of E$_a$ with the increasing carbon chain length as shown by Castañeda et al. (2012) for

aliphatic aldehydes up to 5 carbon atoms, it is expected to see somewhat lower barriers for similar H-abstractions in hexanal. They calculated the barriers at the CCSD(T)/6-311++G**//BHandHLYP/6-311++G** level. However, using our method the barriers we calculated are roughly 0.5-2.5 kcal mol$^{-1}$ higher in comparison to the values obtained by Castañeda et al. for the different H-abstractions.

It is apparent that our calculated overall rate coefficient $2.3 \times 10^{-12}$ cm$^3$ s$^{-1}$ is about one order of magnitude lower than the

experimental rate coefficient $2.9 \times 10^{-11}$ cm$^3$ s$^{-1}$ measured for the hexanal + OH reaction (Albaladejo et al., 2002; D'Anna et al., 2001) as well as SAR prediction rate coefficient (see Table 1; detailed SAR calculations in Supplement section S1) (Jenkin et al., 2018; Ziemann and Atkinson, 2012). Reducing the barrier height by 1 kcal mol$^{-1}$ (within the error margin of the method used), we obtain the overall rate coefficient $1.3 \times 10^{-11}$ cm$^3$ s$^{-1}$ that is compatible with the reported experimental results.

The potential energy surface (PES) of hexanal for the different H-abstraction channels is shown in Fig. 3. The C1 channel associated with aldehydic H-abstraction has a submerged barrier. Among other channels, the $\gamma$ H-abstraction on C4 shows the lowest barrier. The radical (C$_6$H$_{11}$O) formed via the $\alpha$ H-abstraction is more stable than those obtained from $\beta$, $\gamma$, $\delta$ and primary H-abstractions because the unpaired electron in the former can delocalize over the carbonyl double bond. Although the stability of C$_6$H$_{11}$O radicals with respect to $\beta$, $\gamma$, and $\delta$ H abstractions are similar, the stability of the

radical after primary H-abstraction on C6 is the lowest.

C1 is the dominant channel with a yield of 92% (see Table 1), followed by the C4 channel with a yield of around 4%. The energies of the intermediate peroxy radicals A and D (see Fig. 3; C1 and C4 channels) both with molecular formula C$_6$H$_{11}$O$_3$ are below the separated reactants hexanal + OH by 63.3 and 54.5 kcal mol$^{-1}$, respectively.

### 3.2 H-shift rates

Mechanistic details of the HOM formation through both C1 and C4 channels are shown in Fig. 4. The PES of two lowest energy barrier H-shift pathways of the peroxy radicals A and D are shown in Fig. 5, and the calculated transition state energies and rate coefficients are shown in Table 2. Although we report both MC-TST and MESMER rate coefficients, we use the MC-TST rate coefficients in the following discussions to compare with relevant literature rate coefficients. We found that the MC-TST rate coefficients are roughly one order of magnitude lower than those derived from MESMER. The MC-TST treatment includes the

influence of several lowest energy conformers, while MESMER only considers the lowest energy TS and reactant conformers,





(a) C1 channel

(b) C4 channel

**Figure 4.** Mechanism of hexanal + OH reaction initiated by H-abstraction on (a) carbon C1 and (b) carbon C4. The branching of A between green and purple channels is 91%:9%. A61 branches towards A62 by only 0.1%, while the remaining 99.9% is towards the formation of A61a. The branching of D between green and purple channels is 95:5. Since the subsequent branching of A62 towards the dead-end red channel is only 0.3%, the green channel is likely to continue functionalization towards A61a. In the C4 purple channel, D51 branches towards the blue route by 3%, still giving the same product as the parent purple channel.






**Figure 5.** Stationary points along the PES of hexanal OH oxidation reaction (continuation from Fig. 3). Zero-point corrected energies are shown on the y-axis and the reaction coordinate on the x-axis. TS = transition state, Hscr = H-scrambling. Labels with a prime, e.g., A61′, indicate alkyl radicals.





**Table 2.** Calculated rate coefficients for H-migration in peroxy radicals. The migrating H-atoms are marked in red.

| Reactant | Product | Substitution pattern | | Span[a] | Barrier | $k_{MC-TST}$ (s$^{-1}$) | $k_{MESMER}$ (s$^{-1}$) | $\Delta ZPE$ |
|---|---|---|---|---|---|---|---|---|
| | | H-atom | –OO | | | | | kcal mol$^{-1}$ |
| A | A61′ | –CH$_2$– | –C(=O)OO | 1,6 | 19.1 | $1.69 \times 10^{-1}$ | 2.0 | 2.5 |
| A | A51′ | –CH$_2$– | –C(=O)OO | 1,5 | 20.5 | $3.49 \times 10^{-2}$ | $1.97 \times 10^{-1}$ | 2.9 |
| A61 | A61a′ | –CH$_2$– | >CHOO | 1,5 | 21.5 | $3.90 \times 10^{-3}$ | $2.05 \times 10^{-2}$ | 7.8 |
| A61 | A62 | –C(=O)OOH | >CHOO | 1,8$^\dagger$ | 25.3 | $3.72 \times 10^{-6}$ | $1.33 \times 10^{-5}$ | 9.5 |
| A62 | A62a6′ | –CH(OOH)– | –C(=O)OO | 1,6 | 18.5 | 2.08 | – | – |
| A51 | A51a6′ | –CH$_3$– | >CHOO | 1,6 | 24.4 | $2.77 \times 10^{-5}$ | $2.19 \times 10^{-4}$ | 13.8 |
| A51 | A51a5′ | –CH$_2$– | >CHOO | 1,5 | 22.3 | $7.81 \times 10^{-4}$ | $8.57 \times 10^{-3}$ | 11.0 |
| D | D61′ | –C(=O)H | >CHOO | 1,6 | 18.7 | $8.63 \times 10^{-1}$ | 20.43 | 2.9 |
| D | D51′ | –CH$_2$– | >CHOO | 1,5 | 21.1 | $3.91 \times 10^{-2}$ | 1.08 | 6.2 |
| A62 | A61 | –CH(OOH)– | –C(=O)OO | 1,8$^\dagger$ | 15.8 | $6.92 \times 10^{2}$ | $7.53 \times 10^{2}$ | -9.5 |
| D51 | D52 | –CH(OOH)– | >CHOO | 1,7$^\dagger$ | 17.2 | $8.96 \times 10^{1}$ | $2.41 \times 10^{2}$ | 1.2 |
| D52 | D52n′ | –C(=O)H | >CHOO | 1,6 | 19.1 | $1.38 \times 10^{-1}$ | 5.15 | 1.8 |
| D51 | D52n′ | –C(=O)H | >CHOO | 1,4 | 23.1 | $2.67 \times 10^{-2}$ | $4.43 \times 10^{-2}$ | 3.0 |

$^a$ H-shift span, $^\dagger$H-scrambling reactions

and therefore the estimates are likely the upper limits. Both sets of rate coefficients are provided in Table 2 to establish a range of possible values.

The initial abstraction of the H atom from C1 by OH creates a radical center on the terminal carbon giving intermediate A′ (Fig. 4a). The intermediate A′ has the possibility to either fragment towards pentyl (C$_5$H$_{11}$) radical and carbon monoxide (CO)

or add an O$_2$ molecule to form an APR intermediate A (C$_6$H$_{11}$O$_3$). The calculated rate coefficient of the fragmentation pathway is $2.27 \times 10^3$ s$^{-1}$, which is significantly slower than O$_2$ addition ($\sim 10^7$ s$^{-1}$ under atmospheric conditions). The formation of intermediate A is therefore the most competitive route available to A′. Subsequent intramolecular H-shift reactions of A are key for rapid autoxidation and the formation of highly functionalized products.

In the current analysis, we exclude RO$_2$ + RO$_2$/HO$_2$ and RO$_2$ + NO$_x$ reactions and focus on unimolecular autoxidation

reactions that lead to rapid functionalization and formation of HOM. The intermediate APR A can either follow 1,6-H-shift reaction (C1 green channel) or 1,5-H-shift reaction (C1 purple channel) and subsequently form the peroxy radical intermediates A61 and A51, respectively, both with the composition C$_6$H$_{11}$O$_5$ and having a terminal peroxy acid group. The calculated barrier heights of A towards 1,6-H-shift (green route) and 1,5-H-shift (purple route) reactions are 19.1 kcal mol$^{-1}$ and 20.5 kcal mol$^{-1}$, respectively, with corresponding rates of $1.69 \times 10^{-1}$ s$^{-1}$ and $3.49 \times 10^{-2}$ s$^{-1}$, respectively. We did not find

similar acyl peroxy H-shift rate coefficients reported in the literature for comparison.

The initial H-abstraction by OH from the C4 carbon and the subsequent rapid O$_2$ addition gives the peroxy radical D (Fig. 4b). This peroxy radical can readily abstract the aldehydic hydrogen via an intramolecular 1,6-H-shift reaction and subsequently



form the APR A62 (C4 green channel). The peroxy radical D can also abstract an H atom from carbon C2, which is the next most competitive channel (purple arrows, C4 channel) after the aldehydic H-shift. The calculated barrier heights of the 1,6-H-shift (green route) and 1,5-H-shift (purple route) reactions are 18.7 kcal mol$^{-1}$ and 21.1 kcal mol$^{-1}$ respectively, with corresponding rate coefficients of $8.63 \times 10^{-1}$ s$^{-1}$ and $3.91 \times 10^{-2}$ s$^{-1}$, respectively. The literature rate coefficient of a similar aldehydic 1,6-H-shift from CHO to >CHOO reported by Vereecken and Nozière (2020) is higher ($k = 7.9$ s$^{-1}$) by about a factor of 9 than the rate reported here (1,6-H-shift in C4 channel mentioned above), while the rate coefficient of peroxy H-shift from –CH$_2$– with 1,5 span is about an order of magnitude lower ($k = 1.23 \times 10^{-3}$ s$^{-1}$). The higher rate coefficient for the peroxy H-shift reaction we report can be due to the presence of the $\alpha$-CHO group next to the –CH$_2$– group. The branching of A and D (both C$_6$H$_{11}$O$_3$) towards green and purple sub-channels are governed by these initial H-shift barriers. The MESMER simulation derived branching ratio of A towards green and purple channels are 91% and 9%, respectively, while the same for D are 95% and 5%, respectively.

In the C1 green channel, the alkyl peroxy radical intermediate A61 (C$_6$H$_{11}$O$_5$) with a terminal peroxy acid group needs to overcome a barrier of 21.5 kcal mol$^{-1}$ to undergo a 1,5-H-shift reaction to form a carbon-centered radical on C2. The rate coefficient for this isomerization is $3.90 \times 10^{-3}$ s$^{-1}$. Subsequently, the alkyl radical A61a′ (see Fig. 5, not shown in Fig. 4) can add an O$_2$ molecule to form the C$_6$H$_{11}$O$_7$ peroxy radical (A61a). As the peroxy acid group is significantly more stable than the acyl peroxy group, the barrier for the H-scrambling reaction (between $\gamma$-OO and peroxy acid groups) of A61 is quite high ($E_a = 25.28$ kcal mol$^{-1}$, $k = 3.72 \times 10^{-6}$ s$^{-1}$) making the formation of A62 unlikely (orange double arrow in Fig. 4a).

Along the C1 purple channel, the intermediate A51 needs to overcome the barriers of 22.3 and 24.4 kcal mol$^{-1}$ to undergo 1,5- and 1,6-H-shift reactions to yield A51a5′ and A51a6′ radicals, respectively (see Fig. 5) and subsequently, by O$_2$ addition, they form the non-terminal peroxy radical A51a5 and the terminal peroxy radical A51a6, respectively (purple and blue arrows, Fig. 4a). The corresponding rate coefficients are $7.81 \times 10^{-4}$ and $2.77 \times 10^{-5}$ s$^{-1}$, respectively; too slow to significantly compete with bimolecular RO$_2$ + RO$_2$/HO$_2$/NO reactions. Comparing with the literature data, analogous rate coefficient of peroxy 1,6-H-shift reaction from a terminal CH$_3$ group found by Vereecken and Nozière (2020) is even lower ($k = 4.06 \times 10^{-6}$ s$^{-1}$). The difference is likely due to the dissimilarity in the molecular structures in terms of the functional groups on the carbon chain in the studied cases.

In the C4 green channel, the A62 peroxy radical can immediately undergo an H-scrambling reaction between $\gamma$-OOH and acyl peroxy groups, and form the peroxy radical A61 with a terminal peroxy acid group (the same intermediate as in C1 channel). The fast H-scrambling reaction with a barrier of 15.8 kcal mol$^{-1}$ and rate coefficient $6.92 \times 10^2$ s$^{-1}$ switches the radical center in C$_6$H$_{11}$O$_5$ intermediate turning A62 to more stable A61. This fast reaction is the reverse of the previous A61 to A62 conversion reaction in C1 channel. This rate coefficient is similar to that reported by Vereecken and Nozière (2020) for H-scrambling reactions with 1,8 spans ($k = 4.17 \times 10^2$ to $1.36 \times 10^3$ s$^{-1}$). The $\alpha$-OOH H-atom of A62 can also migrate to the terminal acyl peroxy radical by 1,6-H-shift reaction leading it towards the radical termination channel (red arrows), giving the closed-shell ketohydroperoxide product C$_6$H$_{10}$O$_4$ (see Fig. 4b). Rate coefficient of such 1,6-H-migration to a tertiary peroxy radical was estimated by Vereecken and Nozière (2020) to be $2.45 \times 10^{-2}$ s$^{-1}$. In our acyl peroxy radical case, we found a very fast rate coefficient (2.08 s$^{-1}$) for A62 to A62a6′ conversion. Based on our calculated rate coefficients, the branching ratios





of A62 towards the green and red channel are 99.70% and 0.30% respectively. This channel thus regenerates the same A61 intermediate that follows the identical pathway, as we observed in C1 green channel, leading to the $C_6H_{11}O_7$ peroxy radical.

Continuing to the C4 purple channel, we get to the intermediate D51 with an OOH functionality at C4 and with an intact aldehydic functionality. The D51 radical intermediate can then follow a rapid H-scrambling reaction to form the peroxy radical D52. The H-scrambling rate corresponding to the conversion of intermediate D51 to D52 is around 5 times lower than that of in the green channel. The H-scrambling barrier in the latter case is 1.27 kcal mol$^{-1}$ higher and corresponding D52 intermediate is 1.17 kcal mol$^{-1}$ less stable than D51. However, the faster aldehydic 1,6-H-shift reaction ($k=1.38 \times 10^{-1}$ s$^{-1}$) in D52 makes

it possible to quickly form $C_6H_{11}O_7$ APR through the formation of D52n′ (shown in Fig. 5) acyl radical intermediate followed by $O_2$ addition. The same D52n′ intermediate can be formed also directly from D51 by aldehydic 1,4-H-shift reaction (blue arrow, Fig. 4b) which seems to be very unlikely due to the higher barrier of 23.1 kcal mol$^{-1}$ and corresponding low rate coefficient of $2.67 \times 10^{-2}$ s$^{-1}$. Vereecken and Nozière (2020) reported a similar rate coefficient ($k = 6.64 \times 10^{-2}$ s$^{-1}$) for such aldehydic H-shift reaction with 1,4 span. The APR D52n is likely to follow similar subsequent reactions as the other APR A62.

However, the former has an additional OOH functionality on carbon C2, allowing for two competing H-scrambling reactions. Here the MESMER simulation derived branching ratio of D51 ($C_6H_{11}O_5$) towards purple and blue channel are 97% and 3% respectively. We did not calculate the relative energies of the final $C_6H_{11}O_7$ intermediates which we expect to energetically get further lowered by 25-35 kcal mol$^{-1}$ due to $O_2$ addition at the last step of their formation.

  The overall maximum yields of the different oxidation products up to $C_6H_{11}O_7$ HOM from initial hexanal OH oxidation

are calculated by multiplying the branching ratios of each intermediate along the reaction pathway. When considering the two competing isomerization channels of the APR A shown in Fig. 4a (and excluding possible bimolecular reactions), the overall maximum yield of $C_6H_{11}O_5$ intermediate (A61) through the C1 green channel is 84%, which subsequently leads to an $O_7$ HOM via a 1,5-H-shift reaction. The overall yield of the same A61 intermediate formed by the C4 green channel is 3.8%, which can follow similar chain propagation steps towards HOM. The $O_5$-intermediate associated with the C1 purple channel has a yield of 8.2%, but the subsequent H-shift rates are very slow and it is unlikely to efficiently form an $O_7$ HOM through

this channel. On the other hand, the overall $C_6H_{11}O_7$ yield by the C4 purple channel is 0.2%, where the only limiting step is the initial branching of the green and purple pathways.

### 3.3 Flow reactor experiments

The evolution of mass spectra at variable reaction time experiments is in agreement with the proposed OH initiated hexanal

autoxidation mechanism. In our experiments, we use high precursor concentrations, and the experimental condition does not exclude the bimolecular $RO_2$ + $RO_2/HO_2$ reactions. The formation of $O_7$ HOM ($C_6H_{11}O_7$) is observed at a 3.1 seconds reaction time, strongly supporting the proposed computational mechanism (see panel (b) and (c) of Fig. 6; blue and purple). Besides, kinetic modelling simulations using atmospherically relevant concentrations and including $RO_2$ + $RO_2$ and $RO_2$ + NO loss processes, show the formation of $O_5$ and $O_7$ products in the expected time-scales (details in Section 3.4). The $C_6H_{11}O_6$

peak in panel (a) of Fig. 6 appears as early as 1.4 s reaction time and it almost certainly involves $RO_2$ bimolecular reactions, as only odd numbers of O-atoms would be expected for purely $RO_2$ mediated aldehyde autoxidation. A possible formation



**Figure 6.** Nitrate chemical ionization mass spectra at different hexanal + OH reaction times: Panel (a) 1.4 s, panel (b) 3.1 s, and panel (c) 12 s. The oxidation products are detected as adducts with $NO_3^-$, which is excluded from the labels. All spectra are hexanal background subtracted resulting in several negative peaks in panels (a) and, (b). Panel (d) shows the mass shifts of the product peaks during H/D exchange experiment. TME = tetramethylethylene. Accretion product $C_9H_{16}O_7$ is linked with the TME derived peroxy radical $C_3H_5O_3$.



mechanism of $C_6H_{11}O_6$ peroxy radical is the conversion of $C_6H_{11}O_5$ (A61 in Fig. 4) to an alkoxy radical $C_6H_{11}O_4$ via a bimolecular step, followed by a 1,4-H-shift reaction and subsequent $O_2$ addition (see Fig. S1). The absence of the $C_6H_{11}O_5$ peroxy intermediate can be due to the poor detection sensitivity of $NO_3^-$ for species with less than two hydrogen bonding

functional groups (Hyttinen et al., 2015). In panel (b) of Fig. 6, we observe the appearance of $O_7$ HOM ($C_6H_{11}O_7$), HOM accretion products ($C_{12}H_{22}O_{9-10}$), $O_4$ and $O_5$ closed shell products, as well as a cross reaction product $C_9H_{16}O_7$ (i.e., a peroxide accretion product formed from hexanal derived $C_6H_{11}O_6$ and TME produced $C_3H_5O_3$ peroxy radical; see Fig. S3) at 3.1 s of reaction time. For the closed shell $C_6H_{10}O_5$ product, we propose that it can form through the $C_6H_{11}O_7$ (A61a in Fig. 4) involving a bimolecular step to produce an alkoxy radical $C_6H_{11}O_6$ followed by a 1,4-H-shift reaction and a subsequent

OH loss (see Fig. S2). As the reaction proceeds up to 12 s (panel (c) of Fig. 6; purple), we see that the previously observed product peaks grow larger and more peaks appear, including HOM accretion product $C_{12}H_{22}O_{11}$. The likely formation process of $C_9H_{16}O_7$ and HOM accretion products ($C_{12}H_{22}O_{9-11}$) are discussed in the Supplement section S2.

The structures that we propose for $C_6H_{11}O_7$ (A61a), $C_6H_{11}O_6$, $C_6H_{10}O_5$ contain two labile hydrogen atoms (either two –OOH groups or one –OH together with an –OOH group). Accordingly, the $D_2O$ experiment associated with panel (d) spectrum

(red) in Fig. 6 shows mass shifts of two units for these products in support of the assigned structures. Regarding the accretion products, the mass shifts of 3-4 units in $D_2O$ experiment are according to the presence of 3-4 labile hydrogen containing groups in their structures, and in full agreement with the proposed peroxy radical structures forming them according to a general reaction $RO_2 + RO_2 \rightarrow ROOR + O_2$ (Bianchi et al., 2019; Valiev et al., 2019; Hasan et al., 2020).

## 3.4  Kinetic modelling and atmospheric implications

The hexanal peroxy radical autoxidation process producing HOM described in the current work is in competition with bi-molecular sink reactions under most atmospheric conditions. In our kinetic modelling simulations, we include $RO_2 + RO_2$ and $RO_2 + NO$ reactions as bimolecular sinks to illustrate the relative role of autoxidation in hexanal OH oxidation leading to HOM. We use a range of precursor concentrations to simulate situations from pristine boreal forest conditions to moderately polluted urban conditions (details in Supplement section S3). Considering a cleaner environment with 0.1 ppb of NO

($2.46 \times 10^9$ molecules cm$^{-3}$), a generic $1.0 \times 10^8$ molecules cm$^{-3}$ of $RO_2$, 1 ppb ($2.46 \times 10^{10}$ molecules cm$^{-3}$) of hexanal and $1.0 \times 10^7$ molecules cm$^{-3}$ of OH, the simulation leads to an appreciable concentration ($3.0 \times 10^3$ molecules cm$^{-3}$) of $O_5$-intermediate formed as early as 0.3 s of reaction time while $O_7$ HOM at 3.8 s. The concentrations grow to a maximum of $1.9 \times 10^6$ and $3.2 \times 10^4$ molecules cm$^{-3}$ for $O_5$-intermediate and $O_7$ HOM, respectively, after 10 s of reaction time. Under high hexanal concentrations (maximum of 8.8 ppb), the above product concentrations increase roughly by a factor of 3 to 4.

At a very low NO condition (0.01 ppb), these product concentrations increase by a factor of up to 1.6. Interestingly, even at 1 ppb of NO with $1.0 \times 10^8$ molecules cm$^{-3}$ of $RO_2$, the $O_7$ HOM concentration goes as high as $1.3 \times 10^4$ molecules cm$^{-3}$ after 10 s of reaction time. However, a NO level of 4 ppb practically halts $O_7$-HOM production through C4 channel even at higher hexanal condition, while 40 ppb of NO significantly prohibits any $O_5$-intermdeiate formation. Although we use higher precursor and oxidant concentrations in our laboratory experiments (1 ppm hexanal, 225 ppb ozone, 40 ppb TME; to allow for

efficient HOM production and detection), the trend of evolution of mass spectral peaks match with that of the simulations. The



H-shift rate coefficients in autoxidation are highly temperature dependent and the MESMER simulations carried over a range of temperatures show that the second H-shift rate coefficients which lead to $O_7$ HOM are increased by a factor of ~2 to 8 at 310 K and 330 K, respectively, relative to the rate coefficients at 298.15K (see Supplement section S4). It should also be noted that bimolecular reactions involving NO, $RO_2$ and $HO_2$ that produce RO radicals do not necessarily terminate autoxidation but
can also propagate it (Iyer et al., 2018; Rissanen, 2018; Wang et al., 2021; Newland et al., 2021; Mehra et al., 2021)

## 4    Conclusions

This work illustrates how a common aldehyde, hexanal, has the potential to rapidly autoxidize to a HOM, and thus contribute to atmospheric secondary organic aerosol budget. Our results on the initial H-abstraction channels of hexanal by OH are consistent with the previous literature results. Abstraction of the aldehydic H-atom is the most competitive pathway, and it
leads primarily to an acyl peroxy radical as the competing CO loss channel is slow relative to $O_2$ addition under atmospheric conditions. Subsequent H-shift reactions of the acyl peroxy radical are surprisingly fast, and lead to a 5-oxygen containing product in sub-second timescales. Apart from the aldehydic H-abstraction channel, the H-abstraction from the C4 carbon by OH is the next most competitive pathway for hexanal. As the subsequent peroxy H-shift of the aldehydic H-atom is rapid, and this abstraction ultimately leads to the same $O_5$ peroxy acid containing $RO_2$ as the most prominent abstraction route. The
following H-shift reaction rate coefficient is slow and bimolecular processes can intervene with the autoxidation chain. On the other hand, although the initial branching of the primary $RO_2$ towards hydroperoxy substituted $RO_2$ with the aldehydic group intact is very small, the following H-shift rate coefficients are significant enough to rapidly form $O_7$ HOM. In the current work, we show how gas-phase autoxidation of aldehydes can be a direct source of condensable material to urban atmospheres even under moderately polluted conditions, and should be accounted for in assessing the air quality and particle loads of any
atmospheric environment.

*Data availability.*    Kinetic modeling results and an example MESMER input file corresponding to one of the studied reactions are provided in the Supplement. The ab initio output files (.log and .out) are available online (https://doi.org/10.5281/zenodo.6560506).

*Author contributions.*    MR, SB, and SI devised the research. SB carried out the electronic structure and master equation calculations, and kinetic modelling simulations with contributions from SI and PS. MR, SB, and AK designed the experimental setup, SB and AK carried out
the experiments, and SB analyzed the data. SB wrote the paper with contributions from all co-authors.

*Competing interests.*    The authors declare that they have no conflict of interest.



*Acknowledgements.* This project has received funding from the European Research Council under the European Union's Horizon 2020 research and innovation programme under Grant No. 101002728 (ERC Consolidator Grant Project ADAPT). The support from the Academy of Finland (331207 and 336531), and Doctoral school of the Faculty of Engineering and Natural Sciences of Tampere University are greatly appreciated. We thank the CSC IT Center for Science in Espoo, Finland, for providing the computing resources.



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
