# Peer review of "An aldehyde as a rapid source of secondary aerosol precursors: Theoretical and experimental study of hexanal autoxidation"

_EGUsphere, 2023_

## Author Comment (AC1)

We thank the reviewers for going through our manuscript titled "*An aldehyde as a rapid source of secondary aerosol precursors: Theoretical and experimental study of hexanal autoxidation*" and for their constructive suggestions on how to improve the work. We have incorporated all the suggestion and have modified the revised manuscript accordingly. Below, the reviewer queries are reproduced in red, followed by detailed point-by-point author responses in blue, and the changes and additions to the revised manuscript and the supplement in purple.

**Reviewer: 1**

Comments

This work details an interesting pathway for HOM formation and involves robust theory and experiment working in tandem which is nice to see. The work is well presented and of good scientific quality. I am not sufficiently qualified to comment any further on the experimental techniques used however broadly speaking the theoretical methods seem approariate given the moderately large number of heavy atoms.

My only queries and comments regard the way the multiconformer approach is presented. I know that in some formulations of multiconformer approaches, corrections are applied to ensure the correct hindered rotor limit but I do not see such corrections in expression 1. Are these included or are the conformer partition functions calculated entirely within the harmonic oscilator aproximation? If the conformer hindered rotors are not accounted for then expression 1 could be achieved simply by putting the seperate conformers into MESMER. Additionally if every conformer is treated as harmonic then you also have the potential issue of overcounting of states when you reach the hindered rotor regime.

**Response:** We thank the reviewer for pointing out the importance of accounting for hindered rotor corrections, and they are correct that no hindered rotor corrections were applied to Eq. (1). We have now run new Gaussian DFT calculations at the ωB97X-D/aug-cc-pVTZ level using the Freq=HinderedRotor keyword to identify the hindered rotors and calculate the corrected partition functions. These corrected partition functions were then used in Eq. (1). Their effect on the calculated rate coefficients is discussed in *Supplement section S3* and in our response to the next question.

It would also be nice to see some consideration of the hindered rotation potentials? do you have these? I do appreaciate that this is quite a large system and since this paper has a large experimental part I do not consider hindered rotors crucial to publication, however if you have not done a hindered rotor treatement then the comparison between a single well model and a multiconformer model is a little missleading since strictly speaking the conformers are equilibrated and the multiconformer model is simply a way of approximating the fully coupled configuration space of each species. Related to this a brief look at your MESMER input shows each species has a symmetry number of 1? By not including hindered rotors im slightly concerned you are not accounting for the three fold periodicity in any methyl rotations for example although correct me if these all cancel out between reactants and TS? In summary the lack of consideration of hindered rotors while potentially pragmatic in this case leads to potential pitfalls. At the least I would ask the authors to clarify some of these points and make

minor adjustments to the text acknowledging some of these issues. Otherwise I am very happy to reccomend publication of this manuscript.

**Response:** As suggested by the reviewer, we have now performed hindered rotor calculations on Gaussian to identify the hindered rotors, the corresponding periodicities and barriers, and the corrected partition functions. In the subsequent MESMER simulation, we used the HinderedRotorQM1D method with the Gaussian derived periodicities and barriers. The harmonic frequencies corresponding to the hindered rotors were removed. Hindered rotor coefficients of all the lowest energy conformers (in an excel file) and one MESMER example file have been added to the Zenodo data archive. We also incorporated the corrected partition functions in Eq. (1) to calculate the MC-TST rate coefficients. All the rate coefficients, before and after hindered rotor treatment, are presented in *Supplementary Table S3*. This resulted in a difference in the rate coefficients by about a factor of 2 in most of the cases.

**Changes to manuscript:** In lines 238–243, "Additionally, hindered rotor calculations were performed to assess the error introduced to the calculated rate coefficients by treating anharmonic low frequency torsions as harmonic oscillators. The hindered rotor treatment was not applied to those reactions where hindered rotor calculations failed for one or more conformers. Therefore, the resultant rate coefficients are not used here for the calculation of branching ratios as they are not suitable for direct comparison. The hindered rotor treatment changed the rate coefficients by about a factor of 2 in most of the cases (see Supplement section S3)."

**Changes to supplement:**

Supplementary Table S3. Calculated rate coefficients for H-migration in peroxy radicals before and after hindered rotor treatment. The migrating H-atoms are marked in red.

| Reactant | Product | Substitution pattern H-atom | Substitution pattern –OO | Span[a] | $k_{MC\text{-}TST}$ (s$^{-1}$) | $k_{MC\text{-}TST}$ (s$^{-1}$) hindered rotor | $k_{MESMER}$ (s$^{-1}$) | $k_{MESMER}$ (s$^{-1}$) hindered rotor |
|---|---|---|---|---|---|---|---|---|
| A | A61′ | –CH$_2$– | –C(=O)OO | 1,6 | $1.69 \times 10^{-1}$ | $6.65 \times 10^{-2\,c}$ | 2.0 | $5.30 \times 10^{-1}$ |
| A | A51′ | –CH$_2$– | –C(=O)OO | 1,5 | $3.49 \times 10^{-2}$ | $4.48 \times 10^{-2\,c}$ | $1.97 \times 10^{-1}$ | $8.70 \times 10^{-2}$ |
| A61 | A61a′ | –CH$_2$– | >CHOO | 1,5 | $3.90 \times 10^{-3}$ | $1.95 \times 10^{-3\,d}$ | $2.05 \times 10^{-2}$ | – |
| A61 | A62 | –C(=O)OOH | >CHOO | 1,8$^\ddagger$ | $3.72 \times 10^{-6}$ | $3.12 \times 10^{-7}$ | $1.33 \times 10^{-5}$ | $7.32 \times 10^{-6}$ |
| A62 | A62a6′ | –CH(OOH)– | –C(=O)OO | 1,6 | 2.08 | $1.03 \times 10^{-1}$ | – | – |
| A51 | A51a6′ | –CH$_3$ | >CHOO | 1,6 | $2.77 \times 10^{-5}$ | $1.47 \times 10^{-5\,c}$ | $2.19 \times 10^{-4}$ | $1.32 \times 10^{-4}$ |
| A51 | A51a5′ | –CH$_2$– | >CHOO | 1,5 | $7.81 \times 10^{-4}$ | $3.15 \times 10^{-4\,c}$ | $8.57 \times 10^{-3}$ | $6.12 \times 10^{-3}$ |
| D | D61′ | –C(=O)H | >CHOO | 1,6 | $8.63 \times 10^{-1}$ | $4.25 \times 10^{-2\,c}$ | 4.04 | $3.58^b$ |
| D | D51′ | –CH$_2$– | >CHOO | 1,5 | $3.91 \times 10^{-2}$ | $2.06 \times 10^{-3\,e}$ | $2.15 \times 10^{-1}$ | – |

| | | | | | | | | |
|---|---|---|---|---|---|---|---|---|
| A62 | A61 | –CH(OOH)– | –C(=O)OO | $1,8^{\ddagger}$ | $6.92 \times 10^2$ | $2.58 \times 10^1$ | $7.53 \times 10^2$ | $2.89 \times 10^2$ |
| D51 | D52 | –CH(OOH)– | >CHOO | $1,7^{\ddagger}$ | $8.96 \times 10^1$ | $7.03 \times 10^1$ | $2.41 \times 10^2$ | $4.00 \times 10^2$ |
| D52 | D52n′ | –C(=O)H | >CHOO | 1,6 | $1.38 \times 10^{-1}$ | $1.45 \times 10^{-2f}$ | 5.15 | – |
| D51 | D52n′ | –C(=O)H | >CHOO | 1,4 | $2.67 \times 10^{-2}$ | $9.75 \times 10^{-2g}$ | $4.43 \times 10^{-2}$ | $8.50 \times 10^{-2b}$ |

[a] H-shift span, [‡] H-scrambling reactions, [b] hindered rotor calculation failed for the product conformer, [c] hindered rotor calculation failed for 1 reactant conformer, [d] hindered rotor calculations failed for 4 TS conformers, [e] hindered rotor calculations failed for 1 reactant and 1 TS conformer, [f] hindered rotor calculations failed for 5 reactant and 2 TS conformers, [g] hindered rotor calculation failed for 1 TS conformer.

Hindered rotor treatment was not applied to MESMER rate coefficients for the reactions, A61→A61a', D→D51', and D52→D52n', as Gaussian hindered rotor calculations failed for their lowest energy TSs. For the reactions, D→D61', and D51→D52n', hindered rotor calculations were not achieved for their corresponding products. In the MESMER input file, for the reactions with a failed product hindered rotor calculation, we used the product information without hindered rotor potential as this was tested to have a minimal effect in the rate coefficient (e.g., a factor of 1.1 overestimation tested for A→A61' reaction).

Table S3 shows that the hindered rotor treatment either decreased or increased the MESMER rate coefficients within a factor of 2 except for the reaction of A→A61' which decreased by a factor of 3.8. In the MC-TST rate coefficients, where the hindered rotor calculations failed for one or more conformers, we used the original partition functions for those conformers instead. The resultant MC-TST rate coefficients with hindered rotor treatment either decreased or increased by a factor of around 2 to 3 except for the reaction of D52→A52n' where hindered rotor calculations failed for five reactant and two TS conformers and decreased the MC-TST rate by a factor of 9.5.

**Reviewer: 2**

Overall comment:

This work studied the autoxidation kinetics and mechanism of hexanal+OH oxidation through quantum chemical calculation and flowtube oxidation experiments. The calculation results suggest that the major RO2s from hexanal + OH could autoxidize at 0.17 and 0.86 s-1 and are estimated to be rapid enough to compete with bimolecular reactions under typical atmospheric conditions. Thus, the authors suggested that hexanal oxidation may be a rapid source of atmospheric SOA. In general, this manuscript is well-written and presents new and interesting results. But I have a few questions and comments and think they should be addressed before this manuscript can be published at ACP.

Detailed comments:

1. Experimental design. The experiments used ozonolysis of TME (C6H12) to generate OH to react with hexanal (C6H12O) and to study the hexanal oxidation products. Using

TME ozonolysis to generate OH is a common approach, but here, it might not be a good idea, considering that both TME and hexanal are C6 compounds. Some TME oxidation products might be misidentified as hexanal products (see a later comment). I believe that the authors need to provide more thorough experimental evidence that the products identified as hexanal + OH products are not from TME.

**Response:** We thank the reviewer for bringing this to our attention. While conducting the experiments, we have been very aware of any background signals originating from, for example, $TME + O_3$, hexanal, hexanal + $O_3$, and hexanal + TME, that can interfere with hexanal + OH oxidation products. All the background spectra are provided in the *Supplement section S4* (see Figure S1 below) which show that the background signals, including those form TME ozonolysis reaction, are distinct from the hexanal oxidation reaction in question. Also, the presented spectra in Figure 6 of the main manuscript are both, hexanal, and $TME + O_3$ (not mentioned earlier) background subtracted. In the revised manuscript, the Figure 6 caption is now modified accordingly.

**Changes to manuscript:** Figure 6 caption, "….. All the spectra are both, TME ozonolysis ($TME + O_3$), and hexanal background subtracted resulting in several negative peaks in panels (a) and, (b). …"

**Changes to supplement:**

**S4: Background spectra in mass spectrometry**

In order to ensure that the hexanal + OH oxidation products shown in Figure 6 of the main manuscript are either distinct or significantly bigger than any background signals, we recorded all the possible background spectra separately. Supplementary Figure S1 clearly shows that the key oxidation products $C_6H_{11}O_{5-7}$, their corresponding closed-shell products $C_6H_{10,12}O_{5-7}$ as well as the accretion products $C_{12}H_{22}O_{9-11}$ are distinct from any background signals originating from $TME + O_3$, hexanal, hexanal + $O_3$, and hexanal + TME experiments except the mass of 226 that matches with $C_6H_{12}O_5$ at unit mass resolution. However, the reported hexanal OH oxidation spectra in the main manuscript (Figure 6) are all relevant background subtracted indicating that the product signal $C_6H_{12}O_5$ (m/z 226) is significant in 3.1 s and 12 s reaction time experiments.

[Figure]

Figure S1: All background spectra (TME + O₃, hexanal, hexanal + O₃, and hexanal + TME) recorded during hexanal OH oxidation experiments. The reaction of TME + O₃ is the source of the oxidant OH.

1. Calculation uncertainties. At Page 10, Line 204-209, the authors compared the calculated hexanal + OH rate constant with prior experimental measurements and suggested a factor of >10 lower in the calculation. But by reducing the barrier height by 1 kcal mol-1, a more consistent result was obtained. This 1 kcal mol-1 was suggested to be the error margin of the calculation method. I wonder if considering this "error margin" in the H-shift process, how much of uncertainty will be estimated for the autoxidation rate constants (i.e., 0.17 and 0.86 s-1)?

**Response:** Considering the error margin of the calculation methods, an uncertainty of 1 kcal $mol^{-1}$ in barrier height leads to about a factor of 5 error margin in the calculated rate coefficients. We now make a note of this in the revised manuscript.

**Changes to manuscript:** In line 309, "Note that the reported unimolecular rate coefficients have an error margin of about a factor of 5 for the method used."

Interpretations of some mass spectral peaks and formation mechanisms should be revised or discussed.

(i) As discussed in section 3.3, a C6H11O6 peak was observed as the dominant product in just 1.4 sec of hexanal oxidation. The author proposed a mechanism of C6H11O5-RO2 + RO2, followed by the formed C6H11O4-RO undergoing autoxidation to produce C6H11O6. If this is really the case, then how can the authors argue that autoxidation outcompetes bimolecular RO2 reactions at short time? It sounded like the short-flowtube method was to make autoxidation chemistry more prominent than bimolecular chemistry. Seeing a bimolecular reaction product in the shortest time seems contradicting. Could this C6H11O6 be from TME + O3? Is it present without hexanal?

**Response:** Up to the formation of $O_5$-$RO_2$ products, the process seemed to be very fast and outcompetes the bimolecular $RO_2$ + $RO_2$ reactions. The following autoxidation processes are found slower, and the bimolecular reactions play a role in parallel to the unimolecular reaction propagation. However, the bimolecular intervention does not stop the autoxidation process but rather decreases its importance due to increasing competition for $RO_2$. So, within 1.4 seconds of reaction time, bimolecular steps are already expected at our reactant concentrations. As provided in the pure TME + $O_3$ oxidation spectrum (see our response above and Figure S1), the dominant hexanal OH oxidation product $C_6H_{11}O_6$ is absent when there is no hexanal flow added to the reactor (*i.e.*, the $C_6H_{11}O_6$*$NO_3^-$ cluster should be seen at 241 Th in the lowest panel of Figure S1).

(ii) Page 17, line 318. The authors proposed that C6H10O5 can be formed from C6H11O7 + RO2 à C6H11O6 + H-shift à C6H10O5 + OH. Why not directly forming C6H10O5 via the classic Russell mechanism: C6H11O6-RO2 + RO2 à ketone + alcohol. It appears that some C6H12O5 is also formed, possibly as the pairing alcohol. This reaction (RO2 + RO2 à ketone + alcohol) appeared to be neglected throughout the entire study (i.e., supporting information S2).

**Response:** According to the Russell mechanism, a representative $RO_2$ (e.g., $C_xH_yO_z$) would produce closed-shell products (e.g, $C_xH_{y-1}O_{z-1}$ (ketone), and $C_xH_{y+1}O_{z-1}$ (alcohol)) with one oxygen atom less than the original $RO_2$ itself. Accordingly, if a simple Russell mechanism is dominant, presumably we would get equal amounts of ketone and alcohol (e.g., $C_6H_{10}O_5$ and $C_6H_{12}O_5$ yielding from $C_6H_{11}O_6$-$RO_2$) from a single $RO_2 + RO_2$ reaction **that we don't see**. This is why we proposed the formation of these closed-shell products rather from separate reactions, with $C_6H_{12}O_5$ being produced from $C_6H_{11}O_5 + HO_2$ reaction, (i.e., $C_6H_{11}O_z + HO_2 = C_6H_{12}O_z + O_2$). The formation of $C_6H_{10}O_5$ product with the expense of $C_6H_{11}O_7$-$RO_2$ also partially explains the smaller intensity of $C_6H_{11}O_7$ along with its relatively slower production. Also, when $C_6H_{11}O_6$-$RO_2$ is a dominant product at 1.4 seconds reaction time, there is no $C_6H_{10}O_5$ and $C_6H_{12}O_5$ products observed from the Russell mechanism.

Nevertheless, due to the reviewer comment, and the importance of the Russel mechanism for the understanding of $RO_2 + RO_2$ reactions in general, we have now acknowledged the reaction in the revised manuscript and also included it in the *Supplement section S5*.

**Changes to manuscript:** In line 334, "For the closed shell $C_6H_{10}O_5$ product, it could be formed through the classical Russell mechanism producing an alcohol and ketone (Russell, 1957; Hasan et al., 2020) (see Fig S3a), or alternatively, it could also form through the $C_6H_{11}O_7$ (A61a in Fig. 4) involving a bimolecular step to produce an alkoxy radical $C_6H_{11}O_6$ followed by a 1,4-H-shift reaction and a subsequent OH loss (see Fig. S3b)."

**Changes to supplement:**

**S5: Bimolecular reaction products**

**$C_6H_{10}O_5$**

(b)

Figure S3: The Russell mechanism producing closed-shell products, an alcohol and a carbonyl compound directly from a single $RO_2 + RO_2$ reaction (a). Formation of $C_6H_{10}O_5$ product likely involves A61a ($C_6H_{11}O_7$) peroxy radical undergoing bimolecular reactions with other peroxy radicals in the gas mixture (b).

(iii) The closed-shell products, $C_6H_{10,12}O_7$ are much smaller than $C_6H_{10,12}O_{5,6}$. How can this be explained?

**Response:** The smaller $C_6H_{10,12}O_7$ product signals compared to $C_6H_{10,12}O_{5,6}$ can be directly related to their formation processes. Because of the slower rate of $C_6H_{11}O_5$ to $C_6H_{11}O_7$ conversion, smaller $C_6H_{11}O_7$-$RO_2$ production is expected which subsequently affects $C_6H_{10,12}O_7$ products.

(iv) The authors suggested that $NO_3^-$ are more sensitive for species with more hydrogen bonding functional groups (-OH and -OOH). How come in the mass spectra, each pairing ketone has higher response than alcohol (e.g., $C_6H_{10}O_x$ vs. $C_6H_{12}O_x$, x=5-7)?

**Response:** We thank the reviewer for bringing this interesting observation into our attention which encouraged us to perform a few select additional experiments. We completely acknowledge that the closed-shell $C_6H_{12}O_x$ (x=5-7) products having one additional H-bonding functional group than the corresponding $C_6H_{10}O_x$ ketones are expected to have better sensitivity when ionized by $NO_3^-$. Although the detection sensitivity of $C_6H_{12}O_x$ products is supposed to be higher, the signal intensity is limited by their overall concentration which appears to be controlled here by $C_6H_{11}O_x + HO_2$ reaction (*please see above our previous answer on inquiry about the Russell mechanism*). We injected a variable concentration of CO to control the $HO_2$ concentration ($CO + OH \xrightarrow{O2} CO_2 + HO_2$) in the reactor to affect $C_6H_{12}O_x$ product formation. The detailed results are shown in *Supplement section S6*. The results show that the $C_6H_{12}O_x$ to $C_6H_{10}O_x$ ratio (i.e., $H_{12}/H_{10}$) increases with the increase of CO concentration in the flow reactor. The original lower $C_6H_{12}O_x$ production and the observation of the new set of experiments again suggest that the closed-shell $C_6H_{12}O_x$ products are likely to form mainly by the $RO_2 + HO_2$ reaction rather than the classical Russell mechanism ($C_6H_{11}O_x + C_6H_{11}O_x = C_6H_{10}O_{x-1} + C_6H_{12}O_{x-1} + O_2$) in the hexanal oxidation case. Nevertheless, the Russell mechanism producing the said compounds has now also been acknowledged.

**Changes to supplement:**

**S6: Hexanal OH oxidation reaction in presence of CO**

[Figure]

Figure S6: Mass spectra showing the key oxidation products ($C_6H_{10-12}O_{5-7}$) with high-resolution peak fitting in hexanal OH oxidation reaction in presence of variable concentrations of CO. TME ozonolysis (TME + $O_3$) is the source of oxidant OH. The left-most panel of the figure presents the spectra under conditions without hexanal (CO + OH) and without CO (Hexanal + OH) added to the flow reactor.

1. The H2O/D2O exchange experiment (Figure 6d) showed some remaining fractions of the peaks at the original masses (no shift). Does this mean that the exchange did not take place for all isomers? Or is this an H2O/D2O exchange efficiency problem. Can you test it using chemical standards?

**Response:** Yes, this is exactly the H/D exchange efficiency problem. The similar case is observed for the reagent ion signals ($HNO_3NO_3^-$ → $DNO_3NO_3^-$ and $(HNO_3)_2NO_3^-$ → $(DNO_3)_2NO_3^-$; see the figure below). Although the complete H/D exchange was not achieved, which can be tedious to obtain even with minor traces of "normal water", $H_2O$, present in the sample, the extent of the exchange was satisfactory to serve the purpose of structural elucidation of our key oxidation products.

**H/D Exchange**

[Figure]

Figure R1 (for review only): Incomplete H/D exchange seen in the reagent ion signals.

1. Kinetic modeling results. For the section 3.4, I feel a figure that summarizes the results of the various modeling scenarios would make it more clear. In addition, how about RO2 + HO2? Is it considered at all? What is the aldehyde photolysis rate constant used in the kinetic model? How does photolysis affect the autoxidation pathway and RO2 fates?

**Response:** In addition to Figure S7 showing how fast HOMs form in a standard condition, a new Figure S8 (see below) summarizing various simulation conditions is added to the *Supplement section S7*.

In this work, as we considered the $RO_2 + HO_2$ reaction as a sink (opposed to autoxidation chain propagation) and did not consider the following chemistry, we did not include the $RO_2 + HO_2$ reaction in the kinetic simulation. In other words, the $HO_2$ is contributing to the oxidation chain termination reaction, and it is considered within the total generic $RO_2$ concentration, i.e., ($HO_2$ is another form of $RO_2$ where R = H).

In general, the reaction with OH radical is the dominant loss process for linear-chain aldehydes, while photolysis competes with OH reaction for the branched-chain aldehydes. (Mellouki et al. *Chemical reviews* **2015**, 115.10, 3984-4014; Mellouki et al. *Chemical Reviews* **2003**, 103.12, 5077-5096). Therefore, we did not include the photolysis rate constants in the simulation.

**Changes to supplement:**

**S7: Kinetic modelling of HOM formation**

[Figure]

Figure S8: Kinetic simulation results showing the distribution of major autoxidation products in OH initiated hexanal oxidation reactions with variable precursor concentrations. The concentrations presented in the bar plots are extracted after 10 s of simulation time.

1. Lastly, I think extending the autoxidation mechanistic study into SOA formation in the title and as a main conclusion is going too far. This work only showed that autoxidation happens during hexanal oxidation, but did not perform any measurements for SOA formation. It could be mentioned as an implication or suggested as future work, but should not be highlighted in the title and first sentence in the Conclusion. If the authors really want to discuss more on SOA formation, I suggest at least make some estimates of the volatilities of the autoxidation products. And more discussion in that regard is needed.

**Response:** While we modified the first sentence of the Conclusion section as given below, we would still like for you to consider keeping the title of the manuscript as is. The hexanal autoxidation system studied here works as a base model for longer chain aldehydes, which will oxidize largely by similar pathways, and based on the very recent results by Wang and co-workers (Wang, Z. et al., Comms. Chem. 2021, 4:18.), will most likely be a source of atmospheric SOA. Even specific shorter chain aldehydes have been previously implied as potentially important sources of secondary aerosol (e.g., Chan, A. W. H., *Atmos. Chem. Phys.* 2010, 10, 7169). HOM monomers and dimers with O/C elemental ratios of 0.7 and above

originating from VOC oxidation were previously shown to be direct sources of SOA (Mikael Ehn et al. *Nature* **2014**, *506*(7489), 476-479.), which is implied here also for the aldehyde oxidation HOM products. Furthermore, the mention in the title is about "aerosol precursors", which is still a rather vague term, and thus likely appropriate here.

**Changes to manuscript:** The first sentence of Conclusion section, "This work illustrates how a common aldehyde, hexanal, has the potential to rapidly autoxidize to a HOM, and thus contribute to the condensable material budget that ultimately grows atmospheric secondary organic aerosol."

**Reviewer: 3**

Comments

This work investigated the autoxidation kinetics and mechanism of hexanal+OH oxidation through state-of-the-art quantum chemical methods and flow reactor experiments. It suggests that hexanal (as a case study for aliphatic aldehydes with more than 5 carbon atoms) could be a source of atmospheric secondary organic aerosol. I find the paper very well written and clear, fitting the scope of ACP and overall employing the correct scientific approaches necessary to perform such a study. However, I should state that I cannot comment on the experimental part of the work, as my field of expertise is theoretical and computational chemistry. And in this respect, my attention was drawn precisely to the kinetic details of the hexanal+OH bimolecular reaction. My comments and questions are the following:

**1)** The authors use robust MC-TST calculations to treat the unimolecular H-shift reactions, but then use a much simpler approach (Eq. (2)) to calculate the rate coefficient for the hexanal+OH reaction. Why is that? And why is there no tunneling correction in Eq. (2)? Simple and cost-effective bimolecular MC-TST protocols that account for these issues have been recently proposed, for example A. S. Petit and J. N. Harvey, PCCP, vol 14, 184-191 (2012) and L. P. Viegas, Environ. Sci.: Atmos. DOI: 10.1039/D2EA00164K [and references therein].

**Response:** We thank the reviewer for emphasizing the use of more reliable TST equations that include tunneling corrections to calculate the H-abstraction rate coefficients in hexanal + OH reactions. We agree that by adopting the suggested approaches the reported rate coefficients will be more reliable. The references provided by the reviewer have been very helpful in our exploratory work. We initially considered our use of Eq. (2) (the simpler approach which is now Eq. (1) in the *Supplement section S2* below) to be sufficient to provide accurate branching ratios as the number of TS conformers and the magnitude of the imaginary frequencies are both generally low for OH H-abstraction reactions. We acknowledge, however, that a more robust approach is necessary especially when comparing our values to those in the literature.

We have now made detailed comparisons of the rate coefficients that were calculated using the three different approaches. The tunneling factor is included in the modified equations. We found that the use of the more robust bimolecular MC-TST equation resulted in a slightly higher rate coefficients except for the aldehydic H-abstraction due to the higher number of reactant hexanal conformers relative to the aldehydic H-abstraction TS (see details in the

*Changes to manuscript* section below). We now discuss the results of the different approaches in the revised manuscript.

**Changes to manuscript:** In lines 146–161, "The rate coefficients of the H-abstraction reactions of hexanal by OH, were calculated in three different ways (see Supplement section S2 for details). The rate coefficients presented here involve only the lowest energy conformers of TS and reactants using the bimolecular TST expression shown in Eq. (2).

$$ k = \sigma\, \kappa\, \frac{k_B T}{h * c^\circ} \frac{Q_{TS}}{Q_{OH}\, Q_{Hex}} exp\left(-\frac{E_{TS}-E_R}{k_B T}\right) \qquad (2) $$

where, $\sigma$ is the symmetry factor that is related to the reaction path degeneracy. Here, the values of $\sigma$ are 1, 2, and 3 for the aldehydic H-abstraction, abstraction from a secondary carbon (C2-C5), and abstraction from the primary carbon (C6), respectively (Castañeda et al., 2012). $c^\circ = p^\circ/k_B T$ is the total concentration of molecules in standard condition, $2.46 \times 10^{19}$ molecules cm$^{-3}$. $Q_{Hex}$ and $Q_{OH}$ are the partition functions of the lowest energy hexanal and OH conformers, respectively.

The equation that accounts for multiple conformers of TS and hexanal (Viegas, 2018; Viegas and Jensen, 2023) significantly reduced the rate coefficient of aldehydic H-abstraction. We only found one TS conformer for this reaction, while we had 18 conformers of hexanal that were within 2 kcal/mol in relative energy. Because the former goes into the numerator and the latter in the denominator of the bimolecular MC-TST equation, this reduced the rate. Viegas and Jensen (2023) studied fluorinated aldehydes with three carbon atoms employing MC-TST equation which likely did not encounter this issue with the TS conformers. Castañeda et al. (2012) studied the similar aldehydic system as we do where they used the lowest energy conformer TST (LC-TST) equation. We therefore adopted the same in this work to calculate the rate coefficients and consequently the branching ratios of OH H-abstractions of hexanal."

**Changes to supplement:**

**S2: Bimolecular TST expressions for H-abstraction reactions – a comparison**

The rate coefficients ($k$) of the H-abstraction reactions of hexanal by OH, were calculated in three different ways. A simpler approach which is based on the reaction free energy barrier ($\Delta G^{\ddagger}$) is given in Eq. (1).

$$ k = \frac{k_B T}{h * c^\circ} exp\left(-\frac{G_{TS}-G_R}{k_B T}\right) \qquad (1) $$

The constants, $k_B$, and $h$ are Boltzmann's constant and Planck's constant, respectively. Absolute temperature, $T$, is set to 298.15 K. $c^\circ$ is the total concentration of molecules in standard condition, $2.46 \times 10^{19}$ molecules cm$^{-3}$. $G_{TS}$ and $G_R$ are the Gibbs free energies (at 298.15 K and 1 atm) of the TS and the reactant, respectively.

The equation that accounts for multiple conformers of TS and hexanal, based on multiconformer transition-state theory (MC-TST) (Møller et al., 2016), is shown in Eq. (2).

$$k = \sigma \, \kappa \, \frac{k_B T}{h * c^{\circ}} \frac{\sum_{i}^{all\,TS\,conf.} \exp\left(-\frac{\Delta E_i}{k_B T}\right) Q_{TS,i}}{Q_{OH} \sum_{j}^{all\,R\,conf.} \exp\left(-\frac{\Delta E_j}{k_B T}\right) Q_{Hex,j}} exp\left(-\frac{E_{TS}-E_R}{k_B T}\right) \quad (2)$$

where, $\sigma$ is the symmetry factor (Castañeda et al., 2012), and $\kappa$ is quantum mechanical tunneling (Henriksen and Hansen, 2018). $\Delta E_i$ is the zero-point-corrected energy of the $i^{th}$ TS conformer relative to the lowest-energy transition state conformer, and $Q_{TS,i}$ is the partition function of the $i^{th}$ transition state conformer. Similarly, $\Delta E_j$ and $Q_{Hex,j}$ are the corresponding values for hexanal conformer $j$. $Q_{OH}$ is the partition function of the lowest energy OH conformer. $E_a = E_{TS}-E_R$ is the zero-point corrected barrier height corresponding to the lowest energy TS and reactant conformers. In the case of only lowest-energy reactant and TS conformers, Eq. (2) is reduced to Eq. (3) as below. The approach is called lowest-conformer TST (LC-TST). The rate coefficients calculated using all the approaches are given in Supplementary Table S2.

$$k = \sigma \, \kappa \, \frac{k_B T}{h * c^{\circ}} \frac{Q_{TS}}{Q_{OH} \, Q_{Hex}} exp\left(-\frac{E_{TS}-E_R}{k_B T}\right) \quad (3)$$

Supplementary Table S2. Overall reaction and TS energies in kcal/mol of the different OH H-abstraction reactions of hexanal along with calculated rate coefficients (in cm$^3$ molecule$^{-1}$ s$^{-1}$).

| H-abstraction channels | $\Delta G^{\neq}$ | $k$ (simple) | $E_a$ | $\kappa$ | $k$ (LC-TST) | $k$ (MC-TST) |
|---|---|---|---|---|---|---|
| C1 (aldehydic H)[‡] | 6.92 | $2.15 \times 10^{-12}$ | -0.03 | 1.0 | $2.13 \times 10^{-12}$ | $4.14 \times 10^{-13}$ |
| C2 ($\alpha$ H) | 9.52 | $2.67 \times 10^{-14}$ | 2.0 | 1.2 | $6.36 \times 10^{-14}$ | $5.01 \times 10^{-14}$ |
| C3 ($\beta$ H) | 9.18 | $4.73 \times 10^{-14}$ | 1.5 | 1.05 | $9.99 \times 10^{-14}$ | $4.50 \times 10^{-14}$ |
| C4 ($\gamma$ H) | 8.79 | $9.14 \times 10^{-14}$ | 0.1 | 1.47 | $2.69 \times 10^{-13}$ | $9.22 \times 10^{-14}$ |
| C5 ($\delta$ H) | 10.28 | $7.39 \times 10^{-15}$ | 1.6 | 1.04 | $1.53 \times 10^{-14}$ | $5.78 \times 10^{-15}$ |
| C6 (primary H) | 10.29 | $7.27 \times 10^{-15}$ | 2.7 | 1.29 | $2.80 \times 10^{-14}$ | $1.39 \times 10^{-14}$ |

[‡]aldehydic H-abstraction barrier calculated at RHF-RCCSD(T)-F12a/VDZ-F12// MN15/def2-tzvp level of theory. $k_{overall\,(simple)}$ = 2.32 x 10$^{-12}$ cm$^3$ molecule$^{-1}$ s$^{-1}$, $k_{overall\,(LC-TST)}$ = 2.61 x 10$^{-12}$ cm$^3$ molecule$^{-1}$ s$^{-1}$.

Table S2 clearly shows that the overall H-abstraction rate coefficient is dominated by the aldehydic H-abstraction channel (C1). Using both approaches, the simple bimolecular TST and LC-TST, the overall rate coefficients are similar with values of 2.32 x 10$^{-12}$ and 2.61 x 10$^{-12}$ cm$^3$ molecule$^{-1}$ s$^{-1}$, respectively, whereas the overall MC-TST rate coefficient is significantly lower because of the aldehydic H-abstraction getting lowered by a factor of around 5. The lower aldehydic H-abstraction rate coefficient in the MC-TST approach is due to the contribution of only one TS conformer against 18 hexanal conformers in the partition function

term of the Eq. (2). The other H-abstraction rate coefficients (C2–C6 channels) show a little increase in both LC-TST and MC-TST approaches compared to the simple bimolecular TST approach except a little decrease in δ H-abstraction (C5 channel) in the MC-TST approach. The γ H-abstraction rate coefficient (C4 channel) in the LC-TST approach according to Eq. (3) shows an increase by a factor of 2 compared to the simple expression in Eq. (1). The tunneling factor $(\kappa)$ in all the H-abstraction cases is close to 1, and therefore, it does not seem to have a significant effect in the rate coefficients. On the other hand, the symmetry factor (σ) shows a significant influence in the rate coefficients except for the aldehydic H-abstraction. Overall, although the relatively more sophisticated bimolecular TST approaches (i.e., LC-TST and MC-TST) do not seem to affect the overall H-abstraction rate coefficient of hexanal, they do have an effect on the non-aldehydic H-abstractions as well as the branching ratios.

**2)** Between lines 100-115 the authors talk about the calculation of TS conformers for this reaction. The authors write "(...)except for the aldehydic H-abstraction, in which case, the initial TS optimization is carried out using the MN15/def2-tzvp level of theory instead of B3LYP/6-31+G* since the latter method failed to find the TS structure. The conformer sampling step on the OH aldehydic H-abstraction TS structures did not lead to additional conformers." I am a bit confused on how many aldehydic H-abstraction TS structures were found. Could the authors clarify this and the total number of TSs for the hexanal+OH reaction? Also, how many conformers for the hexanal reactant were found?

**Response:** In line 111, we could identify a typing error, "The conformer sampling step on the OH aldehydic H-abstraction TS  structure did not lead to additional conformers." This is now corrected in the revised manuscript. In this case, we obtained only one TS conformer while 18 conformers were found for the hexanal reactant that are within 2 kcal/mol in relative energy. Details in the above response.

**3)** Reactions between the OH radical and volatile organic compounds are well known to often proceed via the formation of a pre-reactive complex that precedes the hydrogen abstraction step. The literature is incredibly vast on this. However, Figure 3 does not show the formation of these complexes. Could the authors explain why?

**Response:** In Figure 3, the pre-reactive complex was initially not included just for simplification. The Figure 3 is now modified including the presence of the pre-reactive complex, in the revised manuscript. Please see the modified figure below.

**Changes to manuscript:**

[Figure]

Figure 3. Relative electronic energies (zero-point corrected) of pre-reactive complexes (RC), transition states (TS) and products in different H-abstraction channels of hexanal OH oxidation reaction. The enlarged view of the reaction barriers, $E_a$ ($\Delta ZPE^{\pm}$), relative to separated reactants and without pre-reactive complexes, is presented in the inset of the figure. The potential energy surface (PES) is extended for C1 and C4 channels up to the intermediates A and D respectively ($C_6H_{11}O_3$).

**4)** Between lines 200-210 the authors discuss the quality of their obtained bimolecular rate coefficient, which is lower than the experimental and SAR rate coefficients by approximately one order of magnitude. The authors then state that "Reducing the barrier height by 1 kcal mol -1 (within the error margin of the method used), we obtain the overall rate coefficient 1.3x10(-11) cm3 s-1 that is compatible with the reported experimental results.", based on (and please correct me if I misunderstood this) the fact that their barrier heights are higher than the ones obtained by Castañeda et al. (2012) at the CCSD(T)/6-311++G**//BHandHLYP/6-311++G** level. First of all, I think that barrier heights calculated at the RHF-RCCSD(T)-F12a/VDZ-F12 level over wB97X-D/aug-cc-pVTZ or MN15/def2-tzvp geometries are of better quality than CCSD(T)/6-311++G**//BHandHLYP/6-311++G** calculated barriers. So, in my opinion, the problem with the underestimated rate constant does not come from the barrier heights. Secondly, I do think that this underestimation is being caused by the lack of an MC-TST treatment to this reaction, as well as the lack of a tunneling coefficient in Eq. (2). Also, by showing Eq. (2) in that form, it makes the reader think that the authors only used one reactant and TS conformer for the calculation. However, small aldehydes such as CF3CH2CHO and CF3CF2CHO have at least 7 low energy aldehydic transition states (L. P. Viegas, DOI: 10.1039/D2EA00164K) which strongly contribute to the final rate constant value. Table 1 of the same paper also shows the underestimation effect on the final rate constant by considering

only the lowest energy reactant and TS conformer, k_OH^LC-TST(calc), compared to a more complete MC-TST value, k_OH(calc). So, this multiconformational effect added to a tunneling factor could place the calculated hexanal+OH rate coefficient much closer to the experimental result.

In light of these comments, I ask the authors if they could please clarify these issues and make the necessary adjustments to the manuscript before it can be accepted to be published in ACP.

**Response:** As we have already discussed in the earlier response, using more robust bimolecular TST protocols as suggested by the reviewer generally increased the rate coefficients. However, including multiple conformers lead to lower H-abstraction rate coefficients than the LC-TST protocol as the number of the reactant hexanal conformers were invariably higher than TS conformer, leading to a larger reactant partition functions in the denominator, and consequently lower rate coefficients.

We agree with the reviewer that the computational methods used in this work are of high quality. The discrepancies with reaction rate coefficients found from the literature still remain but are within our computational uncertainties.

**Changes to manuscript:** No additional changes to what was done in response to the reviewer's first comment.

---

## Author Response (AR2)

We thank the reviewers for going through our manuscript titled "*An aldehyde as a rapid source of secondary aerosol precursors: Theoretical and experimental study of hexanal autoxidation*" and for their constructive suggestions on how to improve the work. We have incorporated all the suggestion and have modified the revised manuscript accordingly. Below, the reviewer queries are reproduced in red, followed by detailed point-by-point author responses in blue, and the changes and additions to the revised manuscript and the supplement in purple.

**Referee #2 (report #2)**

The revision addressed most of my earlier concerns. I appreciate that the authors performed additional experiments to verify a few questions. I have just two additional questions regarding their responses.

1. The authors uphold their interpretation of the C6H11O6 peak in 1.4 s reaction time, as from C6H11O5-RO2 + RO2 to form C6H11O4-RO and this RO isomerizes to form C6H11O6. How fast is this RO isomerization to compete with O2 addition and RO decomposition? If the authors' interpretation is true, then why were there no other products from RO2 + RO2 (i.e., C6H12O5 alcohol, which has two O-H bonds). For typical RO2 + RO2 reactions, especially for secondary RO2 as shown in Figure 4, the following reactions are expected to take place at the same time (different branching ratios, but both are major pathways):

RO2 + RO2 = R-OH + R=O (1)

RO2 + RO2 = 2*RO (2)

For that secondary C6H11O5-RO2, the branching ratio to form RO is expected to be ~ 0.6 and the R-OH + R=O to be ~0.4. Some RO may add oxygen to form R=O again and some RO2 may decompose. Thus, I would expect that the C6H11O6-RO2 have similar abundance as C6H10O5+C6H12O5. Although C6H10O5 may not have great sensitivity, the C6H12O5 product should be present in the mass spectrum. Here, what I cannot understand is that the author suggest that reaction (2) is dominant in as short as 1.4 s, but no evidence of reaction (1) was observed in 1.4 s. Why so?

In addition, in their response, the authors claimed that reaction (1) is not important because their observation didn't show similar R-OH and R=O intensities. Isn't this self-contradicting? Or why do the authors think reaction (2) is important, but reaction (1) is not under the same RO2/HO2 condition for the same reacting RO2?

**Response:** First, we acknowledge the fact that the secondary $C_6H_{11}O_5$-$RO_2$ can undergo both reactions (1) and (2). The $C_6H_{11}O_5$-$RO_2$ radical undergoing reaction (1) can form less oxygenated closed-shell products $C_6H_{10}O_4$ (i.e., R=O) and $C_6H_{12}O_4$ (R-OH) but not $C_6H_{10}O_5$+$C_6H_{12}O_5$ as the reviewer pointed. Despite the two H-bonding groups, the target molecule ($C_6H_{12}O_4$) with 4 oxygen atoms is likely too weakly bound to $NO_3^-$ to be detected. (Hyttinen et al. *J. Phys. Chem. A* **2018**, 122, 269−279). In Hyttinen et al. (2018), they show that, for their model systems with two hydrogen bonding functional groups, molecules with 4 oxygen atoms have a binding energy with $NO_3^-$ that is weaker than that of the nitrate dimer

($HNO_3 * NO_3^-$). As a consequence, the $NO_3$-CIMS has a low sensitivity for the 4 oxygen containing molecule ($C_6H_{12}O_4$).

On the other hand, the $C_6H_{11}O_5$-$RO_2$ undergoing reaction (2) can produce a reactive RO with which can follow isomerization reaction (in competition with RO + $O_2$, and RO decomposition reactions) which is the key to form the dominant $C_6H_{11}O_6$-$RO_2$ (an alkyl peroxy radical with even number of oxygen atoms originated from hexanal OH oxidation) observed experimentally. Based on our current understanding, an alkoxy radical with five (or more) carbon atoms will be more prone isomerization than fragmentation.

Now, unlike the secondary $C_6H_{11}O_5$-$RO_2$ undergoing one bimolecular step ($RO_2$ + $RO_2$), we would not expect the dominant $C_6H_{11}O_6$-$RO_2$ undergo a second bimolecular reaction within 1.4 s. This is likely the reason we do not see the evidence of reaction (1) for $C_6H_{11}O_6$-$RO_2$ yielding $C_6H_{10}O_5$ + $C_6H_{12}O_5$ within 1.4 s reaction time.

In connection to the previous response, where the reaction (1) was initially given somewhat less importance (although latter acknowledged in the revised manuscript) while explaining the distribution of the closed-shell $C_6H_{10}O_5$ and $C_6H_{12}O_5$ products in the higher residence time experiments, we agree that the reactions (1) and (2) take place at same time. We thank to the reviewer to point out the branching ratios of reactions (1) and (2).

2. In Figure S1, the chemical formulas of the major peaks should be labeled.

**Response:** The mass spectral peaks in Figure S1 are not identified which we refer as background signals (mainly originating from the zero-air source) and the background spectra are subtracted accordingly from the hexanal OH oxidation spectrum as mentioned in the main manuscript.

**Referee #3 (report #1)**

The authors did a great and thorough job in addressing all reviewers comments. However, there is an aspect that I would like to see clarified with more detail, and it is related to the fact that the authors only find one TS conformer for the aldehydic H-abstraction. I performed a very quick search at the M08-HX/pcseg-2 level and found 5 unique aldehydic TS conformers, all within 1 kcal/mol. I would suggest that the authors take the Cartesian coordinates that I am providing, reoptimize these structures at their chosen level of theory and verify if they obtain more TSs than the one that they already have. If they do, what is the effect of including them on the rate constant?

The Cartesian coordinates:

O -3.5157423078 -0.7705013644 0.1908610098
C 0.7295103392 0.2362464807 0.0888256259
C -0.7031565879 -0.2837666536 0.0508583823
C 1.7640772110 -0.8618431621 -0.1346671071
C -1.7212591969 0.8330119581 0.2192926397
C 3.1928290922 -0.3327974852 -0.0768257030
C -3.1505390214 0.3610514630 0.2295652609
H 0.8584176104 1.0167253695 -0.6737712758

H 0.9166933345 0.7190967625 1.0575299052
H -0.8508818691 -1.0342321772 0.8357511188
H -0.8871268365 -0.7984151944 -0.8991902775
H 1.6260839022 -1.6472611747 0.6186773913
H 1.5819242107 -1.3325066647 -1.1082434133
H -1.5702141044 1.3852218175 1.1574736605
H -1.6418975814 1.5885380611 -0.5736739767
H 3.9255633199 -1.1266264212 -0.2430670777
H 3.4011294573 0.1139902353 0.9001411968
H 3.3535831202 0.4383711427 -0.8369647663
H -3.9030952629 1.2234925022 0.2744898154
O -4.7425391183 2.6142884258 0.0096841856
H -5.6069971016 2.1937664316 -0.0999879850

O -3.4632383150 -1.2695763565 -0.2703158252
C 0.6105894638 0.2476681491 -0.0790420399
C -0.7513137063 -0.4373748733 -0.0715364753
C 1.7609843555 -0.7228303991 0.1694463041
C -1.8913539513 0.5496751621 -0.2724131522
C 3.1197699219 -0.0336832260 0.1191459376
C -3.2488109529 -0.1002675124 -0.3298417846
H 0.7610224543 0.7486449699 -1.0448610812
H 0.6319163776 1.0375717156 0.6843213910
H -0.8965368888 -0.9687748520 0.8757440787
H -0.7912360671 -1.2007746797 -0.8565820846
H 1.6218987196 -1.2004080747 1.1466246155
H 1.7217393352 -1.5270635406 -0.5754967195
H -1.9267407714 1.3040479251 0.5248642297
H -1.7776767150 1.1189960929 -1.2052902624
H 3.9387600391 -0.7331208689 0.3053154628
H 3.1816556469 0.7622779625 0.8680607240
H 3.2865909901 0.4199266122 -0.8625758896
H -4.1218346267 0.6409079110 -0.4465979948
O -5.5904526812 1.2515381406 -0.6407385895
H -5.9995009141 0.3748431890 -0.6666748132

O -1.9921480153 2.9069191243 0.6693248659
C 0.2789688347 0.1281340706 -0.2832717118
C -1.1372759225 -0.3845756312 -0.0411338072
C 1.3500303778 -0.8835201738 0.1097316982
C -2.1970567562 0.6918681934 -0.2968223693
C 2.7569466902 -0.3864625887 -0.2032364375
C -2.1353665183 1.7344774217 0.7877154290
H 0.3930663892 0.3913684656 -1.3429386142
H 0.4392167554 1.0607779049 0.2761785888
H -1.2363711608 -0.7354208392 0.9931840772

H -1.3342019021 -1.2487820919 -0.6841096227
H 1.2600534114 -1.0990488153 1.1806645123
H 1.1648407674 -1.8319275307 -0.4094728579
H -2.0733065727 1.1727988058 -1.2711137791
H -3.2010896516 0.2559370940 -0.2366823265
H 3.5216850792 -1.1032953814 0.1070058069
H 2.9556618835 0.5622828547 0.3046149203
H 2.8764323615 -0.2150756408 -1.2770248389
H -2.2519239182 1.2516950604 1.8271141020
O -2.8986536896 0.0549237891 2.7706161085
H -3.0965242655 0.6239674825 3.5275766248

O -2.4716811712 2.4502996042 -0.9543102759
C 0.1582715052 0.0224159618 0.2085153514
C -1.2085832933 -0.4140471832 0.7277559536
C 1.2304456505 -1.0463999425 0.3895390511
C -2.2216255991 0.7306426241 0.7241380444
C 2.5650844374 -0.6331255746 -0.2202169446
C -2.3952126534 1.3003145765 -0.6618532347
H 0.0761781909 0.2709882896 -0.8583710502
H 0.4693821849 0.9445075936 0.7188660390
H -1.1171702912 -0.7985411927 1.7491902650
H -1.5825041901 -1.2409522608 0.1105876303
H 1.3536538782 -1.2478924236 1.4603728191
H 0.8887855147 -1.9855783955 -0.0622341052
H -3.2114960089 0.3744578332 1.0363728692
H -1.9294177340 1.5474264520 1.3914339115
H 3.3446253638 -1.3773342030 -0.0384589668
H 2.9036709645 0.3207223815 0.1962856657
H 2.4732929355 -0.5043039823 -1.3024328344
H -2.4359045903 0.4907445393 -1.4814819494
O -1.8759890689 -0.2652507417 -2.8048002245
H -1.7748167363 0.5720492963 -3.2803301294

O -4.2459618367 0.2142833881 1.0732261213
C 0.4283235015 0.2432090007 -0.0620283067
C -0.9398217200 -0.3373903932 0.2785382746
C 1.5645248830 -0.7533760431 0.1411404123
C -2.0455001001 0.7205943374 0.1999882256
C 2.9137678801 -0.1775142143 -0.2741968429
C -3.3976076404 0.0674785811 0.2555289588
H 0.4220403251 0.5814264965 -1.1061205816
H 0.6137519293 1.1363731870 0.5498803216
H -0.9216265986 -0.7661158682 1.2872304512
H -1.1711808100 -1.1563749556 -0.4147513252
H 1.5943327918 -1.0509728000 1.1958950192

H 1.3555183676 -1.6662511153 -0.4297827194
H -1.9632056303 1.4616857746 0.9993902483
H -1.9857408552 1.2214357416 -0.7743573816
H 3.7315789214 -0.8795172761 -0.0930823198
H 3.1302044985 0.7425798220 0.2778636159
H 2.9174699355 0.0700169065 -1.3396955324
H -3.5450103150 -0.6499709515 -0.6341237621
O -3.2435934655 -0.9292906710 -2.2343063529
H -4.1756937406 -0.9110694557 -2.4911863982

**Response:** We thank the reviewer for the additional TS conformer geometries for the OH aldehydic H-abstraction. We reoptimized them at the MN15/def2-tzvp level of theory and found two new TS conformers, one of which is the global minima aldehydic H-abstraction TS conformer, lower than what we reported previously. Of the remaining 3 conformers, one was a duplicate and the other two did not optimize to transition states. Including the two additional TS conformers along with the original TS conformer in the bimolecular MC-TST equation gives us a low aldehydic H-abstraction rate coefficient ($8.57 \times 10^{-13}$). While this is slightly faster than our previous MC-TST rate coefficient ($4.14 \times 10^{-13}$), it is still significantly slower than the literature experimental value. The significantly higher number of reactant hexanal conformers likely has a large influence on this slow rate. However, the new TS conformer leads to a perceptible increase in our lowest conformer TST (LC-TST) rate coefficient. We now provide the new LC-TST rate coefficient and the corresponding branching ratios in the main manuscript (Table 1), and provide both LC-TST and MC-TST rate coefficients in the *Supplementary Table S2*.

**Changes to manuscript:** In Table 1, we updated the aldehydic H-abstraction rate coefficient, branching ratios and the overall rate coefficient.

**Changes to supplement:** In Supplementary Table S2, we updated the LC-TST and MC-TST aldehydic H-abstraction rate coefficients.